# GROUNDING LANGUAGE PLANS IN DEMONSTRATIONS THROUGH COUNTERFACTUAL PERTURBATIONS

**Yanwei Wang**,* **Tsun-Hsuan Wang, Jiayuan Mao, Michael Hagenow, Julie Shah**
MIT CSAIL

## ABSTRACT

Grounding the common-sense reasoning of Large Language Models (LLMs) in physical domains remains a pivotal yet unsolved problem for embodied AI. Whereas prior works have focused on leveraging LLMs directly for planning in symbolic spaces, this work uses LLMs to guide the search of task structures and constraints implicit in multi-step demonstrations. Specifically, we borrow from manipulation planning literature the concept of mode families, which group robot configurations by specific motion constraints, to serve as an abstraction layer between the high-level language representations of an LLM and the low-level physical trajectories of a robot. By replaying a few human demonstrations with synthetic perturbations, we generate coverage over the demonstrations' state space with additional successful executions as well as counterfactuals that fail the task. Our explanation-based learning framework trains an end-to-end differentiable neural network to predict successful trajectories from failures and as a by-product learns classifiers that ground low-level states and images in mode families without dense labeling. The learned grounding classifiers can further be used to translate language plans into reactive policies in the physical domain in an interpretable manner. We show our approach improves the interpretability and reactivity of imitation learning through 2D navigation and simulated and real robot manipulation tasks. Website: https://yanweiw.github.io/glide/

## 1 INTRODUCTION

Language models, in particular, pretrained large language models (LLMs) contain a large amount of knowledge about physical interactions in an abstract space. However, a grand open challenge lies in extracting such semantic knowledge and grounding it in physical domains to solve multi-step tasks with embodied agents. Previous methods, given the symbolic and abstract nature of language, primarily focus on leveraging LLMs to propose abstract actions or policies in purely symbolic spaces or on top of manually defined high-level primitive abstractions (Liu et al., 2023; Ahn et al., 2022; Wang et al., 2023). Such approaches inherently require a set of predefined primitive skills and additional toolkits for estimating affordances or feasibility before executing a plan generated by an LLM (Ahn et al., 2022; Lin et al., 2023).

To address this important limitation, in this paper, we consider the problem of grounding plans in abstract language spaces into robot demonstration trajectories, which lie in the low-level robot configuration spaces. Our key idea is that many verbs; such as reach, grasp, and transport; are all grounded on top of mode families that are lower-dimensional manifolds in the configuration space (as in manipulation mechanics, see Mason, 2001; Hauser & Latombe, 2010). Therefore, LLMs can be prompted to describe the multi-step structure of demonstrations in terms of semantic mode abstractions: valid mode transitions describe pre-conditions for mode-based skills, and mode boundaries explicitly encode motion constraints in the physical space that are critical for task success.

Building upon this idea, we propose *Grounding Language in DEmonstrations* (GLiDE, illustrated in Fig. 1), which casts the language grounding problem into two stages: learning to classify current modes from states, and learning mode-specific policies. The main challenge in mode classification is that learning a decision boundary fundamentally requires both positive and negative labeled examples. To avoid having humans exhaustively provide dense mode annotation that covers the entire state space, we propose to systematically perturb demonstrations to generate "counterfactual" trajectories and use a simple "overall" task success predictor as sparse supervision. Intuitively, perturbations to inconsequential parts of a successful replay add unseen state coverage, while perturbations that cause

---

*Corresponding author: `yanwei@mit.edu`

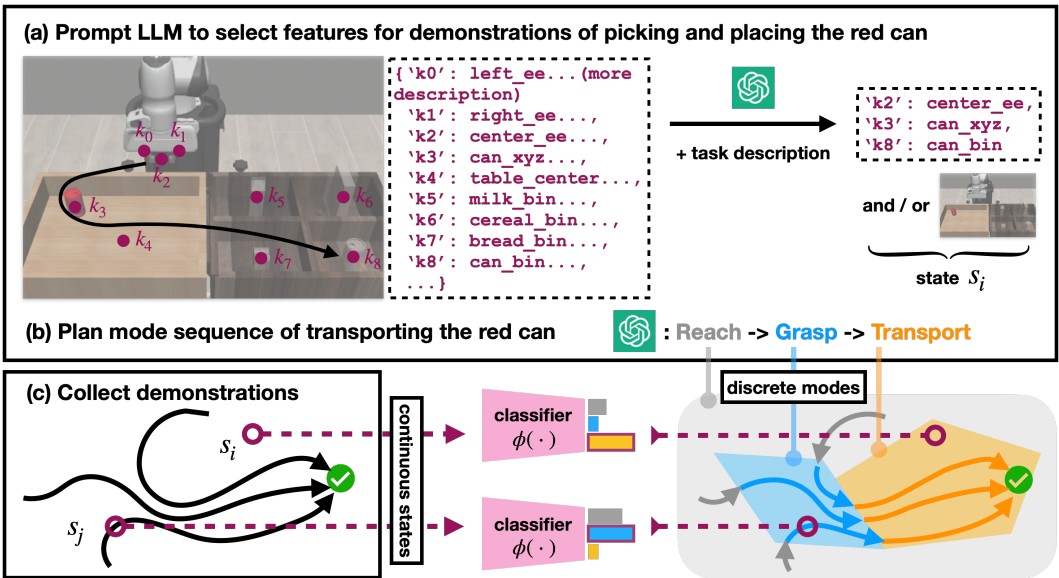

Figure 1: **GLiDE framework** Given a common-sense LLM that understands (a) the appropriate state abstractions for a task and (b) how to solve the task via a sequence of manipulation modes in semantic space and (c) a few unsegmented human demonstrations that embody the transitions through these modes, we learn a grounding classifier that maps continuous physical states and observations to discrete semantic modes. Mode boundaries discovered by the classifier encode constraints implicit in the demonstrations that are critical for task success.

counterfactual failing outcomes reveal constraints in the demonstration. Next, we use an explanation-based learning paradigm (DeJong & Mooney, 1986; Segre & DeJong, 1985) to recover the mode families that successful demonstrations implicitly transition through. With a learned classifier that maps continuous physical states to discrete abstract modes, we can then learn mode-specific policies and also use LLMs to plan for recovery from external perturbations or other sources of partial failures. Our system improves both the interpretability and reactivity of robot learning of multi-step tasks.

Our framework of grounding language plans as recovering modes and learning mode-specific policies brings two important advantages. First, compared to frameworks that generate robot behavior solely based on text, we do not require pre-built policies and feasibility predictors for primitive actions. Experiments show that our learning paradigm can successfully identify each mode from the demonstration data without any human segmentation annotations, and from only a small number of expert-generated demonstrations. Second, connecting demonstrations with language suggests a principled way to improve the interpretability and reactivity of motion imitation. While plenty of data collection systems (Zhao et al., 2023; Fang et al., 2023; Wu et al., 2023; Fu et al., 2024; Chi et al., 2024) allow humans to demonstrate complex multi-step tasks, these demonstrations are typically unsegmented without semantic annotations of individual steps. Neither do humans elaborate on the task constraints that successful trajectories implicitly satisfy. Consequently, the resulting imitation policies cannot detect whether current actions fail to achieve pre-conditions (Garrett et al., 2021) of subsequent actions or replan to recover from mistakes due to covariate shift (Ross et al., 2011). Our system enables the usage of LLMs for replanning and improves the overall system robustness.

## 2 METHOD

Our framework, GLiDE, takes in a language description of the target task, and a small set of successful human demonstrations as input, and aims to produce a robust policy that can accomplish the task successfully even under perturbations. GLiDE first uses a perturbation strategy to augment a small set of human demonstrations with additional successful executions and failing counterfactuals. (Section 2.1). Next, it prompts a large language model (LLM) to decompose the very high-level instruction into a step-by-step abstract plan in language. At this step, the most important outcome is a feasibility matrix that encodes how we can transition between different modes in this task (Section 2.2). Given the augmented demonstration and perturbation dataset and the LLM-generated abstract plan, we ground each mode onto trajectories (Section 2.3) and generate motions for individual modes to be sequenced by a language plan (Section 2.4).

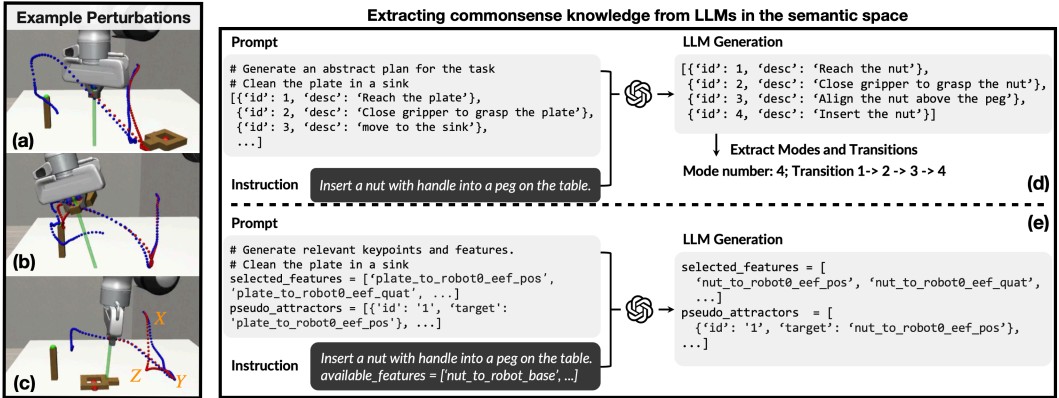

Figure 2: (a-c) Example perturbations causing replays (blue) to deviate from successful demonstrations (red). The task is to pick up the square nut and place it on the peg. End-effector perturbations at different locations (a) may or (b) may not cause grasp failures. (c) The gripper picks up the nut despite an initial end-effector perturbation but later drops it due to a gripper perturbation. LLMs can be prompted (d) to describe a task solution via a discrete mode sequence or (e) to select relevant features and pseudo attractors for solving a task.

## 2.1 DEMONSTRATION DATA AUGMENTATION WITH COUNTERFACTUAL PERTURBATIONS

To learn a grounding classifier that can partition the state space being considered into mode families, we need data coverage beyond the regions explored in a few successful demonstrations. Additionally, to learn mode abstractions that can be used to predict task success–as opposed to clustering data based on statistical similarity–negative data that fail by crossing infeasible boundaries are necessary. Assuming an oracle that can label the execution outcome of a synthetically generated trajectory, we propose the following perturbations to demonstration replays that might reveal task constraints:

**End-effector perturbations** Illustrated in Fig. 2c, given a successful demonstration shown in blue, we first sample two points on the trajectory, namely $X$ and $Y$. Next, we randomly sample a third point $Z$ in the state space. During the replay shown in red, we replace the $XY$ segment with $XZ$ and $ZY$. Depending on the location and magnitude of the perturbations, the robot may still succeed in the task (Fig. 2b) or fail (Fig. 2a), revealing that grasping the square nut is a pre-condition for the next step of peg insertion to be successful.

**Gripper perturbations** Illustrated in Fig. 2c, we randomly toggle the gripper state while otherwise adhering to the original trajectory. Failure replays where the gripper drops the nut pre-maturely reveal the motion constraint of holding the nut during transportation.

Given the perturbed trajectories, we execute them using a trajectory-following controller in the environment and collect a binary task success signal for each trajectory. Essentially, this gives us a dataset of paired trajectories and their task success labels: $\langle \tau^i, succ^i \rangle$, where $\tau^i = \{s_1, s_2, \cdots, s_T\}$, and $succ^i \in \{0, 1\}$. To learn the grounding classifier $\phi(\cdot)$ that can map $\tau^i$ to its corresponding mode sequence $\phi(\tau^i) = \{m_1, m_2, \cdots, m_T\}$ (mode and mode families are used interchangeably in this work), we ask LLMs what modes there are in a demonstration, how they are connected, and what constitutes a state $s_t$ for a given task.

## 2.2 SEMANTIC DESCRIPTION OF DEMONSTRATIONS AND TASK STRUCTURE FROM LLMS

**Explaining continuous demonstrations with a discrete mode sequence** First, we assume a given small set of demonstrations $\{\langle \xi^i, 1 \rangle\}_{i=1}^{D}$, which can be variable at the motion level, satisfy the same sequential transition through $K$ modes, defined as $\sigma \in \Sigma = \{\sigma_i\}_{i=1}^{K}$ and $\phi(s_t) = m_t \in \Sigma$. In other words, if we reduce self-transitions in the demonstrations where $m_t = m_{t+1}$, mode sequence $\phi(\xi^i)$ for all demonstrations can be reduced to the same $K$-step transitions $\sigma_1 \to \sigma_2 \to \cdots \sigma_K$. This is the form of the language plan we prompt LLMs to generate to describe demonstrations. The plan informs the number of modes there are as well as the semantic grounding of each mode as seen in Fig. 2d.

**Representing states with task-informed abstraction** Second, we further prompt LLMs to define the state representation $s_t$ as a set of keypoint-based features or image observations that are relevant to mode classification. In particular, the keypoint-based features come from a pre-defined exhaustive list of keypoints describing the scene as seen in Fig. 1a. Each keypoint definition contains (1) the keypoint name and (2) a short description of its semantic meaning. Given a task description, an LLM

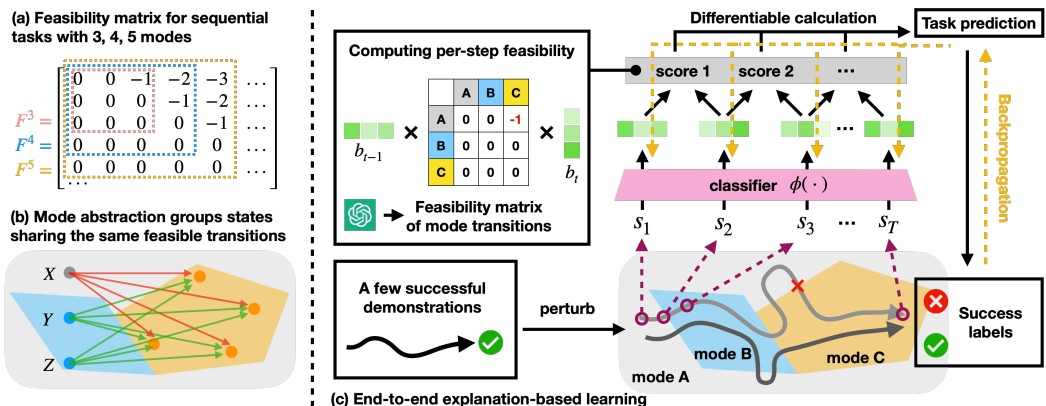

Figure 3: (a) Example feasibility matrices. Specifically, $F^3$ can describe the modal structure for a pick-and-place task with solution reach→grasp→transport, where reach→transport directly is infeasible. (b) The definition of a mode transition implies every state in the second mode is reachable from every state in the first mode (states Y and Z are in the same mode but not X). We leverage this connection between the continuous states and the discrete modes to design (c) a fully-differentiable pipeline that calculates overall trajectory success based on the mode classification of individual states in the trajectory.

can be prompted to either select a subset of keypoints tracking absolute locations or combine pairs of keypoints to track relative positions as shown in Fig. 2e. For image observations, we either use the raw image as a state representation or use a pre-trained vision model (Kirillov et al., 2023) or vision-language model (Huang et al., 2023) to extract features from the image.

**Encoding discrete modal structure in a feasibility matrix** Lastly, while successful demonstrations $\xi^i$ can be reduced to a K-step language plan, not every perturbed trajectory $\tau^i$ can be as it might not be successful or correspond to a minimal solution. Therefore, the reduced mode sequence may contain back-and-forth steps such as $\sigma_1 \to \sigma_2 \to \sigma_1 \to \cdots$ or simply invalid mode transitions. To describe the modal structure of a task in terms of the feasible transitions between modes, we generate a feasibility matrix $F^K$ with $K$ modes by first querying LLMs whether two semantic modes are directly connected. Then we compute the matrix entry $F_{ij}$ from LLM responses as the negative shortest path between each pair of modes. In the case of sequential tasks with a linear temporal structure (true for most experiments considered in this work), zero entries $F_{ij}$ encode valid transitions that incur zero costs. Negative entries $F_{ij}$ encode infeasible transitions, and the magnitudes denote the number of missing modes in between. In particular, in Fig. 3a diagonal entries $F_{ii}$ are feasible self-transitions, and entries $F_{i,i+1}$ are demonstrated mode transitions towards the goal. Note for tasks with complex structures, the matrix may have more negative entries than the ones shown in Fig. 3a. The feasibility matrix is also interpretable and can be modified manually by humans.

## 2.3 END-TO-END EXPLANATION-BASED LEARNING FOR MODE CLASSIFICATION

Given a language plan, a task-informed state representation, and a feasibility matrix as discrete structural information about a task, learning the grounding classifier is an inverse problem that tries to recover the underlying modal structure from sparsely labeled continuous trajectories. To this end, we design a differentiable decision-making pipeline to explain the task success of a trajectory on top of mode predictions. Having trajectory coverage with contrasting execution outcomes allows for recovering the precise grounding in terms of mode boundaries.

**Mode classifier** Our mode classifier is a neural network (with softmax output layers) that inputs a state $s_t$ and outputs a categorical distribution of the abstract mode at that state. Overloading the notation $\phi(\cdot)$ to output both a predicted mode $m_t$ and a mode belief, we have $b_t = \phi(s_t)$. The architecture of the classifier depends on the state representation. The number of softmax categories $K$ is chosen based on the sequence length of the LLM-generated plan. If we had dense mode annotations $\{\langle s_t, m_t \rangle\}_{t=1}^T$, we could train the classifier directly with a cross-entropy loss. However, we only have supervision at the trajectory level via task success. Therefore, we need a differentiable forward model that can predict task success from a sequence of mode beliefs $\{b_t\}_{t=1}^T$.

**Differentiable forward model to predict task success** What makes a perturbed trajectory rollout unsuccessful (or still successful) in solving a task? Following the approach by Wang et al. (2022a),

we consider a successful trajectory as one that both (1) contains only feasible mode transitions according to $F^K$ and (2) eventually reaches the final mode $\sigma_K$ seen in the demonstrations. Since the perturbations we consider in this work do not affect the starting state $s_1$ and final state $s_T$, the success criteria for a trajectory $\tau^+$ solely concerns intermediate transitions: $\phi(s_t^+)F^K\phi(s_{t+1}^+) = 0, \forall s_t^+ \in \tau^+$. Similarly, a failure trajectory $\tau^-$ is one that contains at least one invalid mode transition.

To operationalize this idea, let's consider the dataset of trajectories $\mathcal{T}$ containing both successful trajectories $\mathcal{T}^+ = \{\tau^{i+}\}_{i=1}^M$ and failure trajectories $\mathcal{T}^- = \{\tau^{j-}\}_{j=1}^N$. First, we use a cross-entropy loss to enforce that the starting and ending continuous states for all trajectories must be in the initial and final mode being demonstrated: $\mathcal{L}_{init} = \mathbb{E}_{\tau_i \sim \mathcal{T}} \mathcal{L}_{CE}(\phi(s_1^i), \sigma_1)$ and $\mathcal{L}_{final} = \mathbb{E}_{\tau_i \sim \mathcal{T}} \mathcal{L}_{CE}(\phi(s_T^i), \sigma_K)$. Second, we define the success and failure loss using $f_{t,t+1}$, which is a shorthand for transition feasibility score $\phi(s_t)F^K\phi(s_{t+1})$ between two states:

$$\mathcal{L}_{succ} = -\frac{1}{M}\sum_{\tau_i \in \mathcal{T}^+}\frac{1}{T-1}\sum_{t=1}^{T-1}f_{t,t+1} \qquad \mathcal{L}_{fail} = \frac{1}{N}\sum_{\tau_j \in \mathcal{T}^-}\max(-1, \sum_{t=1}^{T-1}f_{t,t+1}) \qquad (1)$$

Intuitively, minimizing $\mathcal{L}_{succ}$ encourages the classifier to predict mode beliefs such that all transitions between consecutive states are feasible. Minimizing $\mathcal{L}_{fail}$ encourages the classifier to predict mode beliefs such that there exists at least one invalid mode transition. The clipping in $\mathcal{L}_{fail}$ at $-1$ makes the loss well-defined and treats all invalid mode transitions described by the negative entries in Fig. 3a equally[*]. Fig. 3b gives another intuitive example, where states $Y$ and $Z$ constitute the same mode but not state $X$. A necessary condition to test if a state $s$ is in mode $\sigma_i$ is to check if $s$ can directly transition to at least one state in mode $\sigma_{i+1}$ in the trajectory. Adding everything together, we have $\mathcal{L}_{full}$ in Eq. 2, where $\lambda_s$, $\lambda_f$, and $\lambda_i$ are hyperparameters for balancing loss terms:

$$\mathcal{L}_{full} = \lambda_s\mathcal{L}_{succ} + \lambda_f\mathcal{L}_{fail} + \lambda_i(\mathcal{L}_{init} + \mathcal{L}_{final}) \qquad (2)$$

**Extension to underactuated systems.** These conditions are sufficient for recovering modes from a fully-actuated system. For underactuated systems (Tedrake, 2023) such as object manipulation where objects cannot directly move from one configuration to another via teleportation, it is not possible to generate a direct transition between any two modes such as the ones shown in 3b using synthetic perturbations. Hence, we need an additional regularization at the motion level to infer precise boundaries. Specifically, states in the same mode should go through similar dynamics. In other words, one should be able to infer $(s_{t+1} - s_t)$ from $(s_t - s_{t-1})$. Such mapping should be different for different modes. For example, the relative transformation between the end-effector pose and the object pose should remain the same when the robot is rigidly holding the object and change otherwise. Based on this observation, we instantiate a forward dynamics model $\psi(\cdot)$ that inputs the current state change and predicts how the state should change next for each mode. Coupled with a mode belief, we can predict the next state change as $\phi(s_t)^\intercal\psi(s_t - s_{t-1})$. Consequently, we can train mode classifiers for underactuated systems by introducing a dynamics loss $\mathcal{L}_{dyn}$:

$$\mathcal{L}_{under} = \mathcal{L}_{full} + \lambda_d\mathcal{L}_{dyn} \quad \text{where} \quad \mathcal{L}_{dyn} = \sum_{\tau_j \in \mathcal{T}}\sum_{t=1}^{T-1}\|\phi(s_t)^\intercal\psi(s_t - s_{t-1}) - (s_{t+1} - s_t)\|_2^2 \qquad (3)$$

Minimizing $\mathcal{L}_{dyn}$ groups states into modes based on similarity in dynamics. Since losses are differentiable with respect to $\phi$ and $\psi$, we use stochastic gradient descent to optimize learnable parameters.

## 2.4 MODE-BASED MOTION GENERATION

Having learned the explicit mode boundaries, we can leverage them in motion planning to ensure that the robot avoids invalid mode transitions (LaValle, 1998). Alternatively, we can use the classifier $\phi(\cdot)$ to segment demonstrations into mode-specific datasets, with which we can learn imitation policies $\pi_k(a|s)$ for each mode $\sigma_k$ and sequence them using a discrete plan (Wang et al., 2022a). To further improve the robustness of the learned policy for manipulation tasks, we use the mode feature identified by the LLM to construct a pseudo-attractor for each mode. If the mode feature is the absolute pose of the robot end-effector, we compute the mean end-effector poses at which mode transitions $\sigma_k \to \sigma_{k+1}$ occur as the pseudo-attractor; if it is a relative pose, we transform that into an absolute pose of the end-effector at test time. We use this pseudo-attractor to construct a potential field that guides the robot to move towards the next mode at inference time. Specifically, the final

---

[*]Empirically, setting all negative entries in the matrix to be $-1$ can get gradient descent optimization stuck.

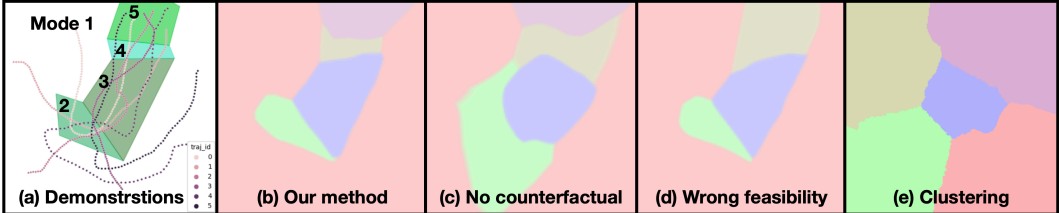

Figure 4: Grounding of 2D navigation task. (a) Given six demonstrations that start in mode 1 and end in mode 5, visualized on top of the ground truth, (b) our method GLiDE recovers the underlying mode abstractions. (c) Without counterfactual data, GLiDE fails to learn precise boundaries. (d) Without a correct feasibility matrix (e.g. 4-mode instead of 5-mode), GLiDE results will miss modes. (e) Lastly, clustering the 2D state space to the nearest mode centers, discovered in the demonstrations by kmeans++, produces an incorrect modal structure.

| Method | 3-Mode | (+perturb) | 4-Mode | (+perturb) | 5-Mode | (+perturb) |
|---|---|---|---|---|---|---|
| Behavior Cloning (BC) | 0.967 | (0.908) | 0.814 | (0.614) | 0.810 | (0.596) |
| GLiDE +BC | 0.963 | (0.887) | 0.892 | (0.753) | 0.893 | (0.753) |
| GLiDE +Planning | **0.996** | **(0.996)** | **0.987** | **(0.966)** | **0.991** | **(0.974)** |

Table 1: 2D navigation success rates for mode-agnostic imitation (BC) and mode-based (GLiDE) imitation or planning for environments consisting of 3, 4, and 5 modes. We test both perturbed and non-perturbed settings 1000 times and report the average success rate. For neural network-based BC policies, we report the average performance across 10 different random seeds.

mode-based policy $\pi_k^*(a|s)$ is a weighted sum of the original $\pi_k(a|s)$ and a control command that moves the end-effector towards the pseudo-attractor for mode $\sigma_k$. We only apply the pseudo-attractor term when the distance between the current state and the pseudo-attractor is greater than a threshold. Intuitively, when a large perturbation leads to out-of-distribution states, the potential field will drive the system back to the demonstration distribution before the imitation policy $\pi_k$ takes sole effect.

## 3 EXPERIMENT

We evaluate our method on three sets of experiments: (1) a 2D navigation task, (2) simulated robot manipulation tasks in RoboSuite (Zhu et al., 2020), and (3) a real-robot implementation of the 2D navigation and a marble-scooping task. Since our robot experiments use end-effector pose control, we refer to task-space features as states rather than configurations.

### 3.1 2D NAVIGATION

**Setup** The 2D navigation environment consists of a sequence of connected randomly generated polygons, and the goal is to traverse from any state in the free space (mode 1) through the polygon sequence consecutively as demonstrated until reaching the final polygon. This environment serves as a 2D abstraction of the modal structure for multi-step manipulation tasks, where each polygon represents a different mode with its boundaries showing the constraint of the mode. Illegal transitions include non-consecutive jumps between modes such as direct transitions from free space to any later modes other than mode 2. This system is fully-actuated with $(x, y)$ coordinate as the state representation and $(\dot{x}, \dot{y})$ as the agent action. For all environments, we use fewer than 10 successful demonstrations for classifier learning and policy learning.

**Results: Mode classification** We visualize the learned grounding classifier in Fig. 4b and Appendix A. Compared to baselines in Fig. 4(c-e), the mode boundaries recovered by GLiDE are the closest to the ground truth shown in Fig. 4a. In particular, the poor grounding learned in Fig. 4(c-e) shows respectively the importance of learning with counterfactual data, a correct task specification from LLMs, and a task prediction loss beyond clustering solely based on statistical similarities in the data. Quantitative results and more visualizations can be found on our website.

**Results: Task execution** Next, we show the learned grounding classifier can be used to improve task success rates, especially in the face of external perturbations. We use behavior cloning (BC) as a mode-agnostic baseline to learn a single policy $\pi(a|s)$ from all successful trajectories. By contrast, our method (GLiDE +BC) first segments the demonstrations and then learns mode-specific

| Method | Can | Lift | Square |
|---|---|---|---|
| GLiDE | **0.83** | **0.83** | **0.67** |
| GLiDE - Dynamics Loss | 0.67 | 0.75 | 0.46 |
| GLiDE - Prediction Loss | 0.67 | 0.68 | 0.56 |
| GLiDE - Feature Selection | 0.55 | 0.70 | 0.57 |
| Traj. Seg. Baseline | 0.66 | 0.56 | 0.54 |

Table 2: Ablation study of the influence of different loss terms and baselines on the mode classification accuracy based on overlap (percentage) with the ground truth.

| Method | Can | Lift | Square |
|---|---|---|---|
| BC | 0.93 | 0.99 | 0.38 |
| BC (p) | 0.20 | 0.18 | 0.03 |
| GLiDE +BC | 0.85 | 0.99 | 0.25 |
| GLiDE +BC (p) | 0.40 | 0.39 | 0.15 |

Table 3: The success rate of mode-agnostic imitation (BC) drops more than that of mode-conditioned imitation (GLiDE +BC) after introducing perturbations (denoted by $p$).

policies. Additionally, instead of mode-based imitation, we could also do planning to stay in the mode boundaries recovered by the classifier, since the system is fully-actuated with a single-integrator dynamics. Specifically, (GLiDE +Planning) uses RRT to compute waypoints to guide motion in non-convex mode 1 and then uses potential fields in convex polygons to generate trajectories that stay in the mode until entering the next mode. Table 1 shows that our methods perform slightly better than BC across different environments. However, when external perturbations are introduced, BC suffers the biggest performance degradation as recovery at the motion level without attention to mode boundaries may incur invalid transitions leading to task failures. The fact that (GLiDE +Planning) can almost maintain the same success rate despite perturbations validates the learned grounding.

**Interpretability**  In the 2D environment, visualization of learned mode families can expose mode constraints and explain why some but not all perturbed demonstration replays fail the task execution.

### 3.2 ROBOSUITE

**Setup**  We test GLiDE across three tasks from Robosuite: placing a can in a bin (*can*), lifting a block (*lift*), and inserting a square nut into a peg (*square*). We use the default action and observation space of each environment unless an LLM suggests different features (e.g., relative distance to an object or keypoints). Since the manipulation tasks define underactuated systems, we use Eq. 3 for training.

**Results: Mode classification**  To evaluate the mode classification accuracy, we manually define the ground truth modes for each environment (details in Appendix B). Table 2 shows the percentage of overlap between mode predictions from different methods and the ground truth mode segmentation. The results show that including all of the loss terms in our method achieves the best boundary alignment with the ground truth as visualized in Appendix B. Ablating the dynamics loss or the task prediction loss ($\mathcal{L}_{succ}$ and $\mathcal{L}_{fail}$) degrades the prediction accuracy as the classifier misses the precise location of important events such as dropping a grasped object. Comparing GLiDE to training without feature selection shows the importance of using an LLM to down-sample the feature space for efficient learning. A trajectory segmentation baseline using kmeans++ clustering on the features also underperforms GLiDE, highlighting the limitation of similarity-based segmentation methods.

**Results: Task execution**  To show the learned grounding can help recover from perturbations, we compare a mode-agnostic BC baseline, which is trained on unsegmented successful demonstrations, to a mode-conditioned method (GLiDE +BC) described in Section 2.4, where each per-model BC policy is augmented with a pseudo-attractor. While our method is insufficient to recover from all potential failures, our goal is to demonstrate how even a basic control strategy that leverages the underlying mode families can benefit policy learning in robotics. Table 3 summarizes the methods' performance without and with perturbations, which will randomly displace the end-effector or open the gripper. We see that for both methods, adding perturbations introduces some amount of performance drop. We find that the performance degradation for the BC baseline is much higher than with GLiDE +BC.

**Interpretability**  In manipulation environments, it is challenging to directly visualize the mode families given the high-dimensional state space. However, exposing the mode families allows us to easily identify mode transition failures which can be used to generate post-hoc explanations of failures (e.g., videos on our website show invalid mode transitions associated with a task failure).

### 3.3 REAL ROBOT EXPERIMENTS: 2D NAVIGATION AND SCOOPING TASKS

**2D navigation**  To illustrate GLiDE can also learn grounding classifiers directly from vision inputs, we implement the simulated 2D navigation task on a real Franka robot, where the end-effector traces through a sequence of colored polygons in a plane. First, we record 20 human demonstrations

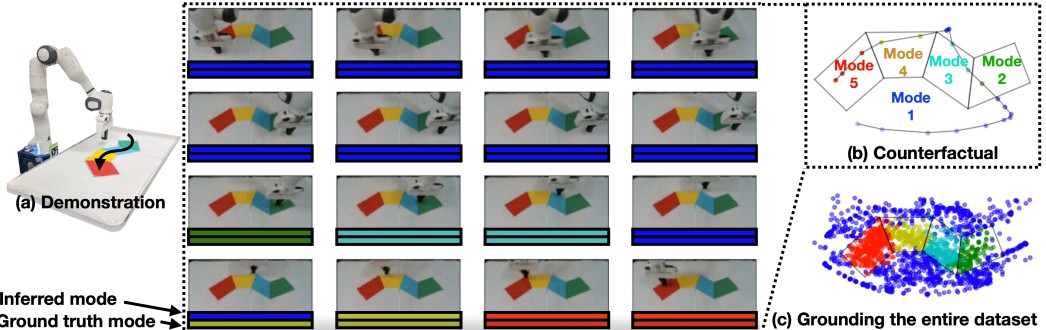

Figure 5: Illustration of the real robot 2D navigation task (a), where the end-effector traces through a sequence of colored polygons. (b) shows a perturbed trajectory, overlaid on ground truth mode boundaries, experiences an invalid transition from mode 1 to mode 4. A vision-based classifier can predict from only pixels the inferred modes (first color bar) that match ground truth (second color bar) with high probability. (c) visualizes the mode prediction of individual image states seen in the dataset. The location of the scattered dots indicates where the images are recorded while the colors show the predictions, which are well-aligned with mode boundaries.

through kinesthetic teaching in the end-effector's state space that start in various parts of mode 1 and end in mode 5 as seen in 5a. Second, we add end-effector perturbations to the demonstration replays to generate coverage over the entire tabletop area and record the perturbed trajectories in image sequences as seen in 5b. For this planar task, we use a simple reset mechanism that brings the end-effector back to one of the demonstrations' starting states after each rollout to collect data continuously. Since we can check if the end-effect is within the convex hull of any colored regions whose vertices are known, we log the mode sequence of each perturbed trajectory and automatically label if the trajectory is a successful task execution by checking if all mode transitions are feasible. Consequently, we were able to collect 2000 labeled trajectories in 2 hours continuously without human supervision. To learn a vision-based classifier, we switch from using multi-layer perceptrons (MLP) for state-space inputs to convolutional neural networks (CNN) to encode image inputs. Figure 5c shows the learned classifier can group image observations into correct modes according to the ground truth mode boundaries.

**Marble scooping task** The second task requires a spoon-holding robot to scoop marbles from a bowl and then transport at least one marble to a second bowl across the table. A typical robot implementation might require engineering a marble detector to check if the spoon is holding marbles and plan actions accordingly (Wang et al., 2022a). Instead, we learn a marble classifier on a wrist camera view to leverage LLM-based replanning as shown in Fig. 6. To collect successful executions, we record the end-effector's pose and wrist camera view as a human demonstrates scooping from various starting states. Since it is non-trivial to engineer a reset mechanism for this task, we ask humans to demonstrate various failures through kinesthetic teaching as well. To improve learning efficiency, we preprocess the raw wrist image to extract a mask corresponding to marble objects, where an empty spoon returns an empty mask. Specifically, we prompt an LLM for relevant object types to track, with which we employ the Segment Anything Model (SAM) (Kirillov et al., 2023) to generate segmentation masks. The classifier is then trained on a state representation consisting of end-effector poses and the marble mask. In Fig. 6, we show how our mode classifier can be used to develop a reactive robot program.

## 4 RELATED WORK

**Learning abstractions from demonstrations.** A large body of work focuses on learning action abstractions from language and interaction. This includes the grounding of natural language (Corona et al., 2021; Andreas et al., 2017; Andreas & Klein, 2015; Jiang et al., 2019; Sharma et al., 2022; Luo et al., 2023), programs (Sun et al., 2020), and linear temporal logic (LTL) formulas (Bradley et al., 2021; Toro Icarte et al., 2018; Tellex et al., 2011). In contrast to learning policies for individual action terms, this paper focuses on learning mode families in robot manipulation domains. These learned mode families enable us to construct robust policies under perturbation (Wang et al., 2022a). Furthermore, our framework is capable of recovering the underlying modes from a small number of *unsegmented* demonstrations. Through the use of synthetic noise (Delaney et al., 2021; Wang et al.,

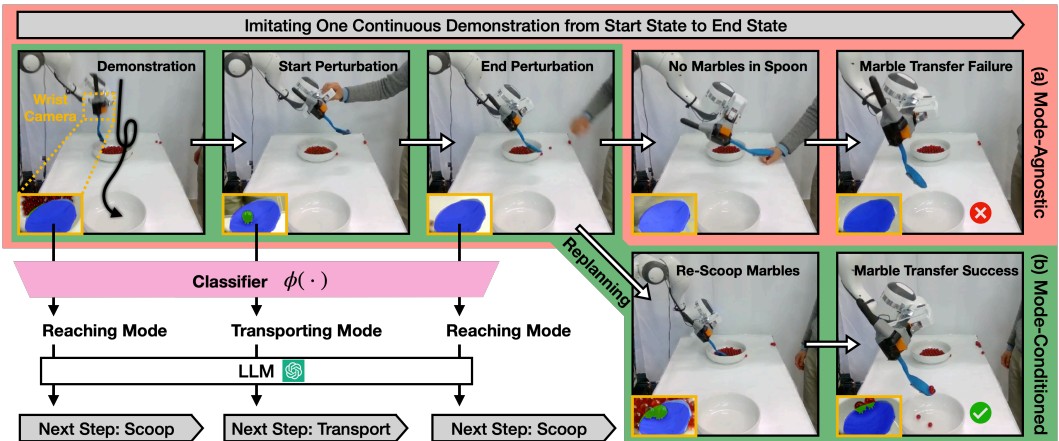

Figure 6: Illustration of the robot scooping task. Here the perturbations are human-initiated (e.g., moving the spoon to drop or fail to grasp marbles). A mode-agnostic BC that imitates continuous motion demonstrations cannot replan at the task level when all marbles are dropped during the transporting mode (a). In contrast, our mode-conditioned policy enabled by the grounding classifier can leverage LLMs to replan given external perturbations (b).

2022b), we relieve humans from the burden of providing negative demonstrations that fail a task by automatically generating positive and negative variations of task executions.

**Grounding language in robot behavior.** With the rise of LLMs that can decompose high-level commands into sequences of actions, there has been much recent interest in the ability to ground these commands in embodied agents. Given that data from environment interactions (including human demonstrations) do not explicitly identify constraints and success criteria of a task, previous work has investigated how to infer affordances directly from observations (Ahn et al., 2022). Compared to prior work (e.g., Lynch et al. (2023)), our method does not require dense labels to learn a grounding operator. We are also not directly using large language models for planning (Huang et al., 2022; Li et al., 2022; Huang et al., 2023). Rather, we are using LLM to guide the discovery of mode abstractions in demonstrations, and as a result, we can also acquire a grounding operator for high-level language commands. In contrast to the discovery of language-conditioned skills (Lynch & Sermanet, 2020; Garg et al., 2022), which can consist of multiple modes, our mode decomposition occurs at a lower level and can explain why certain trajectories fail a task execution.

**Counterfactuals.** Counterfactuals describe hypothetical situations of alternative outcomes compared to the original data (Byrne, 2019). In other words, they are fake (non-human generated) data with a different result (e.g., failing a task instead of succeeding) (Karimi et al., 2020). In this paper, we define counterfactual perturbations as non-human-generated synthetic probes that test which parts of the time-series trajectory data (Delaney et al., 2021) demonstrated by humans have implicit constraints, the violation of which will change the outcome of the successful human demonstrations.

## 5 CONCLUSION

In conclusion, this work introduces a framework, *Grounding Language in DEmonstrations* (GLiDE), to effectively ground the knowledge within large language models into physical domains, via mode families. Given a small number of human demonstrations and task descriptions, we show how GLiDE successfully recovers mode families and their transitions required in the task and enables the learning of robust robot control policies.

**Limitations and future work.** While GLiDE does not need a large number of human demonstrations, it requires a large number of trial-and-errors and an environment with a reset capability in order to collect task success labels of a trajectory. This data inefficiency, however, can be addressed through active learning where the current belief of mode segmentation can be used to probe demonstrations only in regions with high uncertainty. Additionally, prompting the LLM to find a suitable state representation for learning the classifier also requires skill. In future work, we would like to learn the state representation in conjunction with the mode classifiers in an end-to-end fashion.

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

## A  MORE MODE CLASSIFICATION RESULTS FOR 2D NAVIGATION TASKS

Figure 7 visualizes data augmentation of generating additional successful trajectories (middle) and failing counterfactuals (right) from a few successful demonstrations (left). Figure 8 visualizes learned grounding for additional randomly generated 2D navigation environments with a 3-, 4- and 5-mode task structure.

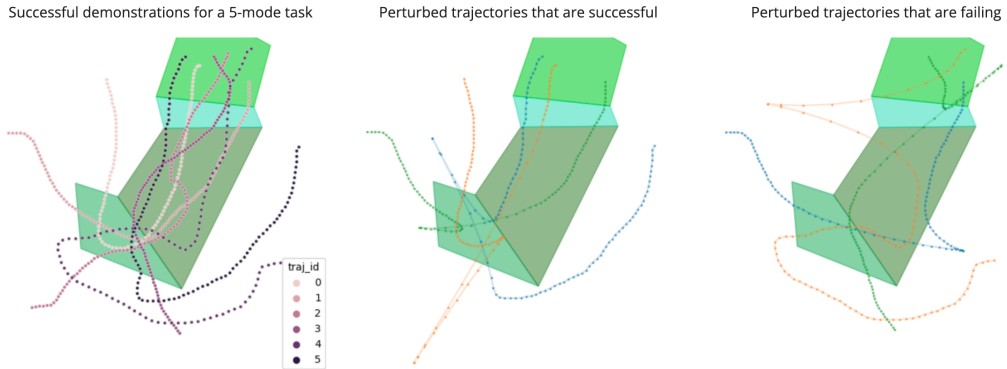

Figure 7: Additional 2D grounding examples. Column (a) shows the ground truth mode segmentation and successful demonstrations. Column (b) shows the learned grounding and its percentage overlap with the ground truth. Columns (c-e) visualize the grounding learned without counterfactual data, a correct feasibility matrix, and task prediction loss respectively.

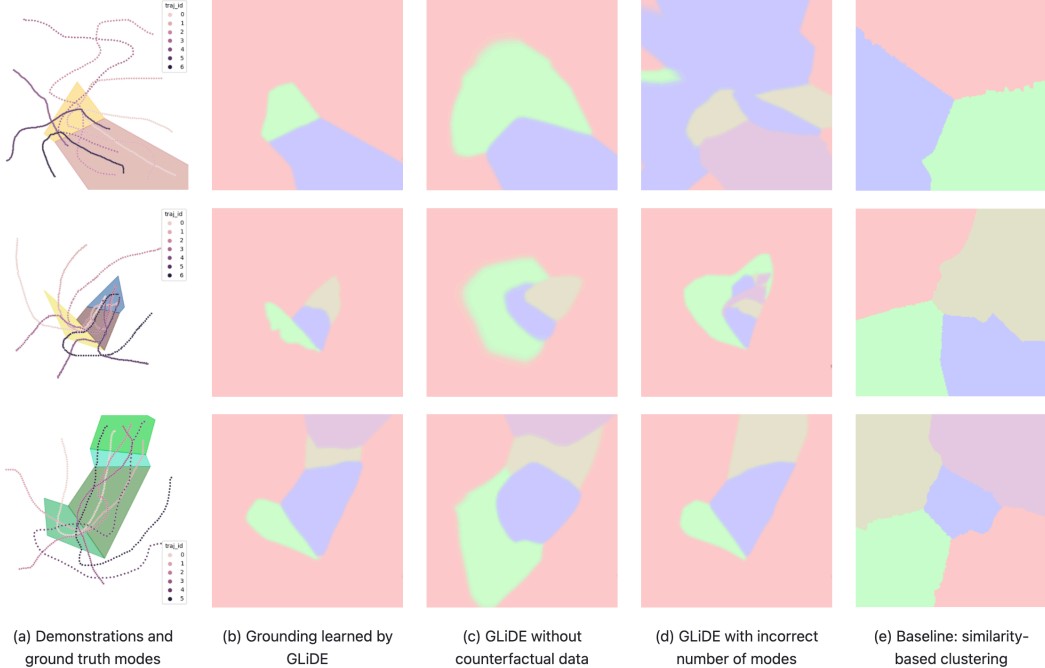

(a) Demonstrations and ground truth modes  (b) Grounding learned by GLiDE  (c) GLiDE without counterfactual data  (d) GLiDE with incorrect number of modes  (e) Baseline: similarity-based clustering

Figure 8: Additional 2D grounding examples. Column (a) shows the ground truth mode segmentation and successful demonstrations. Column (b) shows the learned grounding and its percentage overlap with the ground truth. Columns (c-e) visualize the grounding learned without counterfactual data, a correct feasibility matrix, and task prediction loss respectively.

## B Heuristic Rules for Mode Family Groundtruth in RoboSuite

We use the following heuristic rules to define the ground truth mode families to evaluate the grounding learned by GLiDE:

- *Can (3 modes):* the ground truth modes are reaching for the can (until the end effector makes contact with the can), transporting the can to the bin, and finally hovering about the target bin.
- *Lift (3 modes):* the ground truth modes are reaching for the cube (until the end effector makes contact with the cube), lifting the cube off the table, and finally moving to a certain height above the table.
- *Square (4 modes):* the ground truth modes are reaching for the nut (until the end effector makes contact with the nut), transporting the nut to the peg, aligning the nut above the peg, and finally lowering the nut into the assembled position.

We assess the predicted and ground truth mode (based on the heuristics) for each sample in the robosuite demonstrations to compute accuracy (i.e., the percentage of samples where the predicted and ground-truth modes are the same). Fig. 9 shows the visualization of the mode segmentation from GLiDE, compared to the ground truth on the can placing task. Our model faithfully identifies all the modes and yields a high consistency with the modes defined by human-crafted rules.

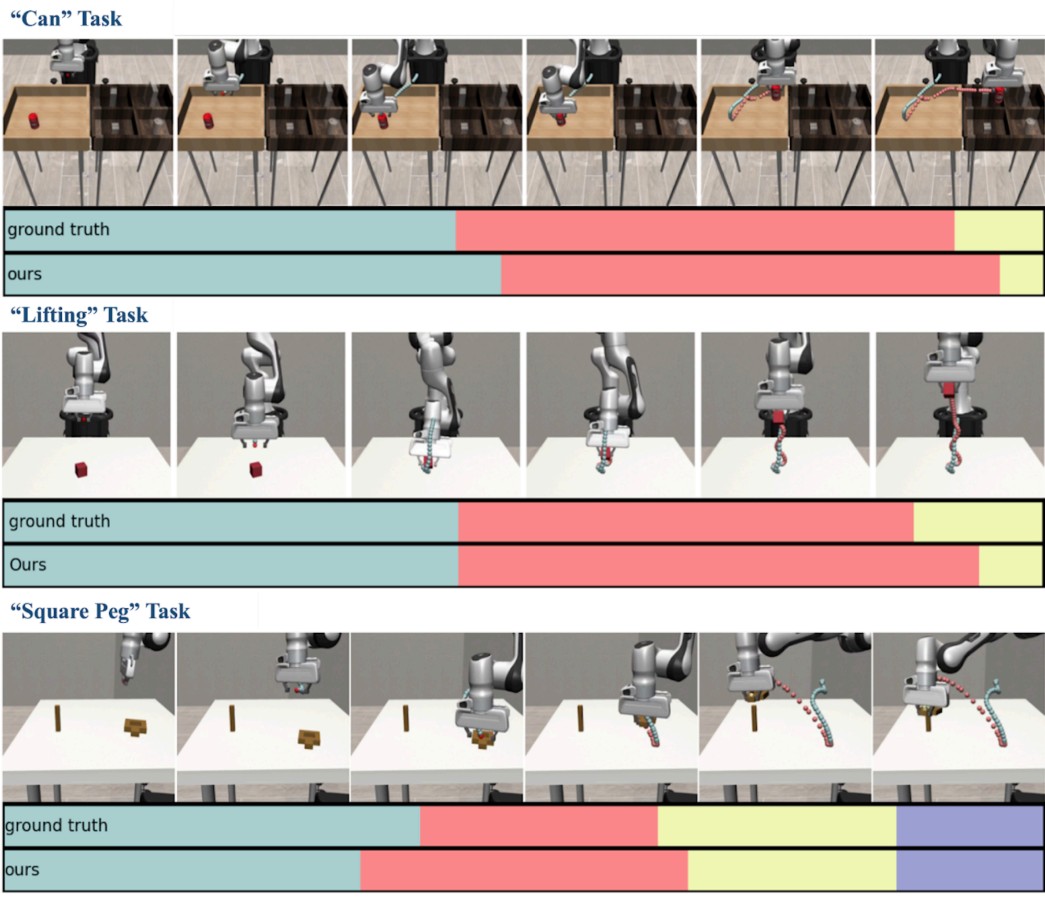

Figure 9: Comparison on the robosuite can task between our method's segmented modes and the ground truth modes. Generally, our system is able to accurately recover mode boundaries.

## C Prompt Engineering For RoboSuite Tasks

### C.1 Prompt Example

```
You are an expert in generating robot action plans and features.

Given a language description of a task, such as "clean the plate in a sink," you should first generate an
    abstract plan for the task. Put them in <plan></plan>. The plan is a list of steps. You should ignore
    object finding or localization actions. For example,
<plan>
steps = [{'id': 1, 'desc': 'Reach the plate'},
 {'id': 2, 'desc': 'Close gripper to grasp the plate'},
 {'id': 3, 'desc': 'Move to the sink'},
 {'id': 4, 'desc': 'Turn on the faucet'}]
</plan>

Next, you should generate a feasibility matrix between all the steps. Put them in <feasibility></feasibility>
<feasibility>
feasibility = {
  (1, 2): True,   # after reaching the plate, we can directly close the gripper
  (1, 3): False,  # after reaching the plate, we can't directly move to the sink
  (1, 4): False,  # after reaching the plate, we can't directly turn on the faucet
  (2, 3): True,   # after closing the gripper, we can directly move to the sink
  (2, 4): False,  # after closing the gripper, we can't directly turn on the faucet
  (3, 4): True,   # after moving to the sink, we can't directly turn on the faucet
}
</feasibility>

The user will also give you a list of available features, such as robot poses, object poses. An example is the
     following:
<available_features>
avaiable_features = ['plate_pos', 'plate_quat', 'plate_to_robot0_eef_pos', 'plate_to_robot0_eef_quat', '
    sink_pos', 'sink_quat', 'sink_to_robot0_eef_pos', 'sink_to_robot0_eef_quat', 'faucet_pos', 'faucet_quat
    ', 'faucet_to_robot0_eef_pos', 'faucet_to_robot0_eef_quat', 'robot0_eef_pos', 'robot0_eef_quat', '
    gripper_state']
</available_features>
We will use the following convention: XXX_pos and XXX_quat denote the pose of the object. XXX_to_robot_eef
    denotes the relative pose between the object and the robot end-effector.

You should select a subset of features. Such as:
<selected_features>
features = [ 'plate_to_robot0_eef_pos', 'plate_to_robot0_eef_quat', 'sink_to_robot0_eef_pos', '
    sink_to_robot0_eef_quat', 'faucet_to_robot0_eef_pos', 'faucet_to_robot0_eef_quat' ,
' 'gripper_state']
</selected_features>

For each step, you also need to generate a "goal"  pose, which is either an absolute pose of the robot or a
    relative pose between the robot and another object. In this case,
<goal>
goals = [
  {'id': '1', 'feature': 'plate_to_robot0_eef_pos'},
  {'id': '2', 'feature': 'plate_to_robot0_eef_pos'},
  {'id': '3', 'feature': 'sink_to_robot0_eef_pos'},
  {'id': '4', 'feature': 'faucet_to_robot0_eef_pos'},
]
</goal>
```

## C.2   LLM RESPONSE FOR CAN TASK

```
Task: move the can from tray 1 to tray 2. Both trays are fixed. Available features:
<available_features>
available_features= ['can_pos', 'can_quat', 'can_to_robot0_eef_pos', 'can_to_robot0_eef_quat', 'robot0_eef_pos
    ', 'robot0_eef_quat',  'gripper_state']
</available_features>

<plan>
steps = [{'id': 1, 'desc': 'Reach the can in tray 1'},
 {'id': 2, 'desc': 'Close gripper to grasp the can'},
 {'id': 3, 'desc': 'Move to tray 2'},
 {'id': 4, 'desc': 'Open gripper to release the can'}]
</plan>

<feasibility>
feasibility = {
  (1, 2): True,   # after reaching the can in tray 1, we can directly close the gripper
  (1, 3): False,  # after reaching the can in tray 1, we can't directly move to tray 2
  (1, 4): False,  # after reaching the can in tray 1, we can't directly open the gripper
  (2, 3): True,   # after closing the gripper, we can directly move to tray 2
  (2, 4): True,   # after closing the gripper, we can directly open the gripper
  (3, 4): False,   # after moving to tray 2, we can't directly open the gripper
}
</feasibility>

<selected_features>
features = ['can_to_robot0_eef_pos', 'can_to_robot0_eef_quat', 'robot0_eef_pos', 'robot0_eef_quat', '
    gripper_state']
</selected_features>

<goal>
goals = [
  {'id': '1', 'feature': 'can_to_robot0_eef_pos'},
  {'id': '2', 'feature': 'can_to_robot0_eef_pos'},
```

```
    {'id': '3', 'feature': 'robot0_eef_pos'},
    {'id': '4', 'feature': 'robot0_eef_pos'},
]
</goal>
```

## C.3 LLM RESPONSE FOR LIFTING TASK

```
Task: lift a block from the table. Available features:
<available_features> available_features= ['block_pos', 'block_quat', 'block_to_robot0_eef_pos', '
    block_to_robot0_eef_quat', 'robot0_eef_pos', 'robot0_eef_quat',  'gripper_state']
</available_features>

<plan>
steps = [{'id': 1, 'desc': 'Move to the block'},
        {'id': 2, 'desc': 'Close gripper to grasp the block'},
        {'id': 3, 'desc': 'Lift the block'}]
</plan>

<feasibility>
feasibility = {
  (1, 2): True,  # after moving to the block, we can directly close the gripper
  (1, 3): False,  # after moving to the block, we can't directly lift the block
  (2, 3): True,  # after closing the gripper, we can directly lift the block
}
</feasibility>

<selected_features>
features = ['block_to_robot0_eef_pos', 'block_to_robot0_eef_quat', 'robot0_eef_pos', 'robot0_eef_quat', '
    gripper_state']
</selected_features>

<goal>
goals = [
  {'id': '1', 'feature': 'block_to_robot0_eef_pos'},
  {'id': '2', 'feature': 'block_to_robot0_eef_pos'},
  {'id': '3', 'feature': 'robot0_eef_pos'},
]
</goal>
```

## C.4 LLM RESPONSE FOR SQUARE PEG TASK

```
Task: insert a nut with a handle to a peg. Available features:
<available_features> available_features= ['nut_pos', 'nut_quat', 'nut_to_robot0_eef_pos', '
    nut_to_robot0_eef_quat', 'handle_pos', 'handle_quat', 'handle_to_robot0_eef_pos', '
    handle_to_robot0_eef_quat', 'peg_pos', 'peg_quat', 'peg_to_robot0_eef_pos', 'peg_to_robot0_eef_quat', '
    robot0_eef_pos', 'robot0_eef_quat',  'gripper_state']
</available_features>

<plan>
steps = [{'id': 1, 'desc': 'Reach the nut'},
 {'id': 2, 'desc': 'Close the gripper to grasp the nut'},
 {'id': 3, 'desc': 'Move to the peg'},
 {'id': 4, 'desc': 'Insert the nut into the peg'}]
</plan>

<feasibility>
feasibility = {
  (1, 2): True,
  (1, 3): False,
  (1, 4): False,
  (2, 3): True,
  (2, 4): False,
  (3, 4): True,
}
</feasibility>

<selected_features>
features = ['nut_to_robot0_eef_pos', 'nut_to_robot0_eef_quat', 'peg_to_robot0_eef_pos', '
    peg_to_robot0_eef_quat', 'gripper_state']
</selected_features>

<goals>
goals = [
  {'id': '1', 'feature': 'nut_to_robot0_eef_pos'},
  {'id': '2', 'feature': 'nut_to_robot0_eef_pos'},
  {'id': '3', 'feature': 'peg_to_robot0_eef_pos'},
  {'id': '4', 'feature': 'peg_to_robot0_eef_pos'},
]
</goals>
```

# D    ADDITIONAL INFORMATION FOR MARBLE SCOOPING TASK

We prompt LLM to generate a subset of features relevant to predicting task success: X, and Y locations of the robot end-effector in the robot base frame as well as the wrist camera mask. Due to a lack of contact sensors, we omit the scooping mode, and prompt the LLM to generate a plan: Reaching → Transporting → Dropping (assuming scooping is always successful when transitioning from the reaching mode to the transporting mode. The corresponding feasibility matrix is $F^3$. In Fig. 10 (top), we plot demonstrations in X and Y and use the color of the scattered plot to indicate ground truth modes (reaching is red, transporting is green, and dropping is blue). Examples of logged spoon masks along these trajectories are shown at the top. At the bottom, we visualize the learned classifier, which has correctly learned three modes (indicated by three distinct colors) by partitioning the space according to X and Y locations and the masks. Note that the location of the learned blue mode matches the dropping bowl location. Figure 11 shows the classifier successfully learns the threshold function that turns the continuous mask values to the discrete information that differentiates reaching mode from transporting mode given the same X, Y locations. Note that for the majority of the wrist camera images we collected, the learned classifier infers the correct ground truth mode.

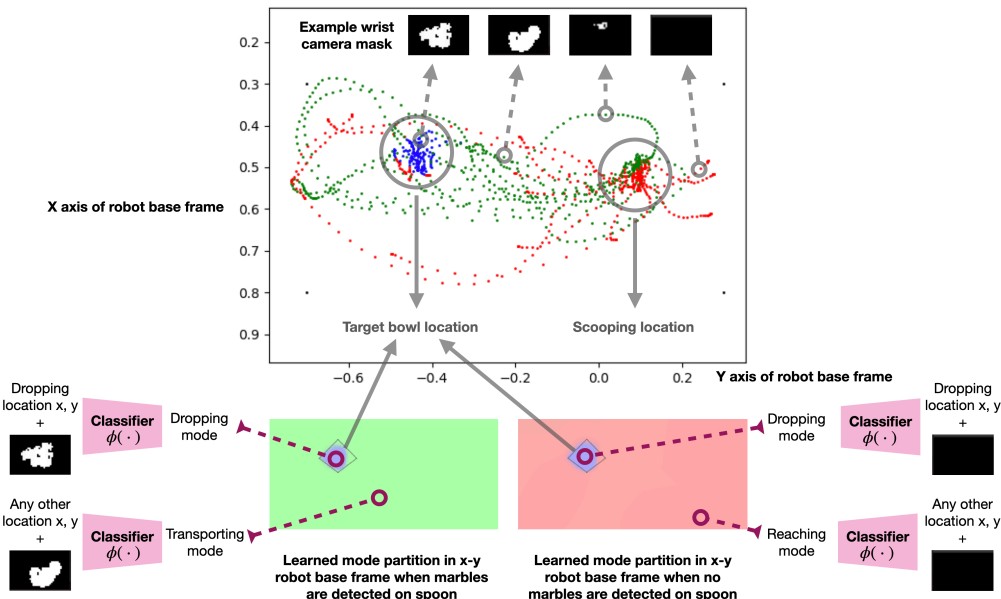

Figure 10: Visualizing the state representation of the scooping task demonstrations and the learned mode partitions.

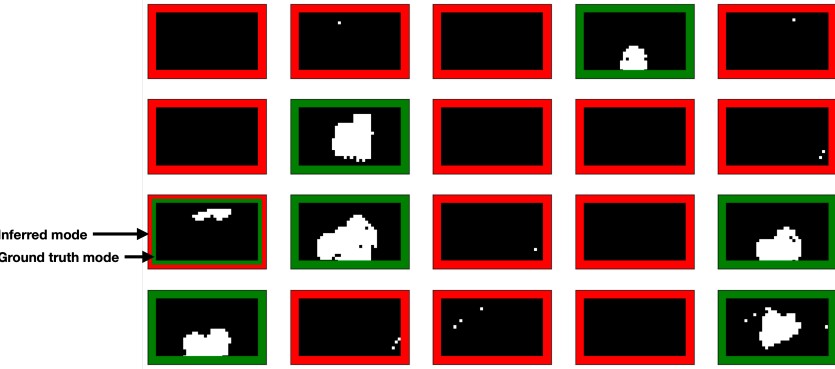

Figure 11: Learned classification over wrist camera images compared to ground truth. The red color indicates "No Marble" and the green color indicates "Has Marble".

