# OpenReview forum: "Grounding Language Plans in Demonstrations Through Counterfactual Perturbations"
_ICLR.cc/2024/Conference — ICLR 2024 spotlight_

### Official Review · Reviewer_i3As · 2023-10-23

**Soundness:** 3 good
**Presentation:** 3 good
**Contribution:** 3 good
**Rating:** 8
**Confidence:** 4

**Summary:**

This paper introduces a framework that autonomously discovers mode families within robot manipulation trajectories with the help of an LLM. Once these mode families are identified, they serve two key purposes: 1) facilitating learning mode-conditioned policies, and 2) enabling the creation of pseudo-attractors that enhance the robustness of the policies against out-of-distribution states.

The LLM plays a vital role in the framework by: 1) decomposing tasks into a sequence of modes, and 2) generating relevant state features for each mode as inputs to the mode classifiers. To train these mode classifiers, the authors propose to augment expert-demonstrated trajectories with negative ones generated by counterfactual perturbations. Additionally, they design several loss terms aimed at aligning the trajectories with the mode sequences generated by the LLM. Specifically, the loss terms encourage: 1) consistent state transitions within the same mode family, and 2) matched mode transitions between the success trajectories and those generated by the LLM.

In a 2D toy example and a robot manipulation domain, quantitative evaluations verify the effectiveness of the proposed method in 1) accurately identifying modes, and 2) learning mode-conditioned policies that are robust against perturbations.

**Strengths:**

This work is a good practice to utilize an LLM to facilitate the learning of motion-level robot - by learning task structures and identifying important state features. The paper is well-written: the problem is sufficiently motivated and the framework is clearly-explained. In the experiments, the authors properly conduct comparisons and ablation studies to demonstrate the effectness of the proposed framework.

**Weaknesses:**

My primary concern of the paper pertains to details of the method, specifically the transition feasibility matrix and transition loss.

Based on the definition of transition feasibility matrix $T_{i,j}=max(i-j+1, 0)$, we get $T_{i,j} = 0$ for $i < j$ and $T_{i,j} > 0$ otherwise. But this doesn’t match the illustration in the right subfigure in Figure 3, where there is “-1” in the matrix. More importantly, the usage of the transition feasibility matrix in the transition loss (in Section 3.2) might be problematic. My guess is that $T_{k,l}$ helps penalize mode transitions following the opposite direction for successful trajectories; however, the successful trajectories seem not to contribute to the loss term at all since $\tau(succ)=0$. I would appreciate more clarifications from the authors on this.

I also think “this work introduces a groundbreaking framework” in the conclusion section is overclaimed. I agree that this is an interesting work, and could be a solid one if the authors properly addess my concern above. However, I would still regard the contribution as incremental, as it only offers an alternative solution to many of the unsupervised trajectory segmentation approach.

**Questions:**

1. When prompting an LLM to generate keypoints, how to make the correspondence between the generated textual description such as “nut-center” and the continuous positions? Do you need to add the list of available keypoints into the prompt?
2. Sometimes the demonstrated trajectories do not perfectly follow the LLM-generated mode transitions. Can the learning framework handle these cases?
3. How much does the LLM-based feature selection contribute to learning the mode classifiers and motion policies? Do you by any chance evaluate the framework using the full state for mode learning?
4. I wonder in the RoboSuite experiment, how do the proposed method (MMLP-Conditional BC) work in the absence of the pseudo-attractor? It will be great to add these results as well as the authors have a better understanding on how different modules contribute to the robustness against perturbations.

---

> ### Author Response · Authors · 2023-11-20
> **Rebuttal (1/3)**
>
> Dear reviewer i3As,
>
> Thanks for your detailed review and feedback! We will first elaborate on our framework and method using two new figures and then respond to your individual questions.
>
> **[What grounding problem are we solving?]** There are at least three kinds of grounding mentioned in the literature: **1. Task grounding** - using language [1] or multi-modal tokens [2] as inputs to an imitation policy to specify tasks/goals. **2. Symbolic grounding** - predicting the Boolean values of symbolic state (e.g. In(can, gripper)=True, On(marbles, spoon)=False, etc) [3, 4, 5] **3. Action grounding** - mapping LLM plan to predefined primitive actions [6, 7, 8] In our work, by grounding we do not mean task grounding, but rather symbolic grounding, where we learn classifiers that map continuous states/observations to discrete modes proposed by LLM. Since we assume each mode is associated with a single policy that’s learned from segmented demonstrations, action grounding can also be achieved as a by-product of learning the classifier, which maps the LLM planned mode sequence to a sequence of policy rollouts.
> We have created a **new figure -- Fig 7** that we include in the revised paper and is currently on the website in [this section](https://sites.google.com/view/grounding-plans/home#h.xly3b8ysna28) that illustrates our overall method including a visual description of how we ground the LLM knowledge in modes.
>
> **[Clarification of our feasibility matrix and transition loss]** We appreciate the reviewer spotting some inconsistency in our original writing and figure 3. In response, we made a new figure to clarify the feasibility matrix and explain the transition loss in terms of success loss and failure loss.
> We have created a **new figure -- Fig 8** that we include in the revised paper and is currently on the website in [this section](https://sites.google.com/view/grounding-plans/home#h.pmzm9p2g4j7v) that illustrates both how we use LLMs to generate the classifier state and how this information is used downstream to compute the classifier loss.
>
> **[How do successful trajectories contribute to the loss term?]** In Figure 8 \(c\), when the classifier is not well trained, it might predict some invalid transitions for successful trajectories and consequently incur a success loss. We plan to update the loss function in the manuscript and update a new version soon.

---

> ### Author Response · Authors · 2023-11-20
> **Rebuttal (2/3)**
>
> **[Is this work merely an alternative to unsupervised trajectory segmentation?]** While we agree with the reviewer that the word "groundbreaking" might be overclaiming, we respectfully disagree our method is merely a trajectory segmentation method. In our work, the capability to segment trajectories is merely a by-product of having learned the grounding classifier. The typical goal of trajectory segmentation in LfD literature is to discover reusable skills that can be activated open-loop [9, 10, 11]. Their goal is not concerned with performance degradation under external perturbations. In contrast, our goal is to learn the boundaries of these mode abstractions that define the valid domains of the discovered skills, as shown in the new Figure 7(d), so that the learned skills can be planned in a closed-loop fashion that is robust to external perturbations. Given the differences in motivations, our framework has two significant technical differences:
> 1. First, since our grounding operator needs to classify/segment not only demonstrated trajectories, but also state space that has not been demonstrated (i.e. need to find mode boundaries), we will need to generate additional data to cover the state space being considered beyond just the demonstration regions. Additionally, in order to learn the boundary we would need executions that succeed by crossing feasible boundaries as well as executions that fail by crossing infeasible boundaries. This additional data generation stage is not considered in typical trajectory segmentation setting that works with only successful demonstrations and not failures. The significantly larger scale of counterfactual data might render non-end-to-end systems such as those using HMM/probabilistic inference impractical [12].
> 2. Second, the aforementioned segmentation methods are not induced by the necessity to predict terminal task failures/success and hence does not necessarily break down demonstrations into minimal abstractions with which planning success can be guaranteed despite perturbations. The key insight is that mode families are a useful construct to help achieve planning success guarantee. The reason that boundary between consecutive mode families have to separate configuration spaces in a particular way is that otherwise the the motion planning won’t guarantee success. Consequently, we can use mode partitions to explain why some but not all perturbations will cause execution failures. Inspired by this idea, we devise a fully differentiable end-to-end explanation pipeline that predicts if a perturbed trajectory is successful or not. Only when the grounding classifier in the pipeline has learned the correct mode partitions, can the overall pipeline differentiating all successful trajectories from failure trajectories. Our explanation-based learning approach is similar to analysis-by-synthesis in other domains. For example, in NeRF only when an accurate 3d representation has been learned can the fully differentiable volumetric rendering pipeline generates images that match groundtruth from all views. To operationalize this idea, our transition loss (both the success loss and failure loss) enforces correct explanation why some perturbations do not fail a successful demonstration but others do, which subsequently enforces our learned classifier to ground continuous boundary states to match those discrete mode families provided by LLM, leading to segmentation of atomic skills useful for replanning under perturbations. To show the importance of our transition loss, we show (1) in 2d polygon domain a clustering-based trajectory segmentation baseline using similarity metric does not lead to correct mode boundaries; and (2) in robosuite domains our loss ablating out the transition loss (effectively an unsupervised trajectory segmentation based only on motion similarity [11]) does not lead to accurate grounding. These new results are documented on an anonymous project website https://sites.google.com/view/grounding-plans/home.

---

> ### Author Response · Authors · 2023-11-20
> **Rebuttal (3/3)**
>
> In particular, in 2d polygon domain, we compare our learned mode classifications with the groundtruth on demonstration trajectories (Therefore the reporting scores are trajectory segmentation accuracies). The following table shows the comparison with ablation model (no counterfactual data) and a simple unsupervised trajectory segmentation baseline by KMeans++ clustering.
>
> | Mode Classifier      | 3-Mode | 4-Mode | 5-Mode |
> | ----------- | ----------- | ----------- | ----------- |
> | Ours      | **0.990**       | **0.967** | **0.970** |
> | No Counterfactual Data   | 0.604  | 0.464 | 0.831 |
> | Trajectory Segmentation Baseline | 0.644 |0.554 | 0.641
>
>
> For robosuite, We also report the average trajectory segmentation accuracy (compared to ground truth) for each method.
>
> | Mode Classifier      | Can | Lift | Square Peg
> | ----------- | ----------- | ----------- | ----------- |
> | Ours (LLM-reduced State Space)     | **0.83**       | **0.83** | **0.67** |
> | Full State Space | 0.55 | 0.70 | 0.57 |
> | Trajectory Segmentation Baseline  | 0.66        | 0.56 | 0.54 |
>
> **[Correspondence between the generated textual description and the continuous positions]** In robosuite environments, the demonstration state consists of predefined object states corresponding to those keypoints shown in figure 8 (a). We add a full list of available keypoints to the prompt when query LLM to find a subset of features relevant to a task. More prompting examples can be found on the website in [this section](https://sites.google.com/view/grounding-plans/home#h.743770ss0gzs).
>
> **[What if the demos do not perfectly follow the LLM generated plan?]** In this work, we give humans LLM-generated plan for humans to provide demonstrations, so that we assume these successful demonstrations can be mapped to the same discrete plan even these demonstrations might come from a multimodal distribution in the continuous configuration space. In future work we will investigate how to map demonstrations with non-unique discrete structure to partial or nonlinear LLM plan.
>
> **[How important is LLM-based feature selection?]** LLM-based feature selection helps improve data efficiency as also shown independently by [13, 14] For example, the state of distractor objects is not useful for learning a classifier that detects different modes in the demonstrations of picking up a can object. Including distractor objects’ states as inputs require significantly more counterfactual data to learn a classifier that does not pay attention to distractor objects. To corroborate this claim, we run the default full set of features of as state representation to learn the grounding classifier for robosuite tasks. We show that the resulting segmentation does not align well with the ground truth (see [table](https://sites.google.com/view/grounding-plans/home#h.7l0jx2g9td4d)) and plan to add qualitative results on the website in [this section](https://sites.google.com/view/grounding-plans/home#h.7l0jx2g9td4d).
>
> **[How MMLP-conditional BC work in the absence of pseudo-attractor?]**
> We apologize for the lack of clarity in the original description. The mode-conditioned policy is conditioned in the sense that the pseudo-attractor varies depending on the predicted mode by the system. However, the imitation policy is not conditioned on mode (i.e., without the pseudo-attractor is the same as the BC). We also tried conditioning the imitation learning on the mode (i.e., learning a different BC network for the state-action pairs classified in each mode), however, we found that this did not substantially impact the performance of the policy.
>
>
> [1] Language Conditioned Imitation Learning over Unstructured Data
>
> [2] VIMA: General Robot Manipulation with Multimodal Prompts
>
> [3] Grounding Predicates through Actions
>
> [4] From Skills to Symbols: Learning Symbolic Representations for Abstract High-Level Planning
>
> [5] Learning Temporal Logic Formulas from Suboptimal Demonstrations: Theory and Experiments
>
> [6] SayCan: Grounding Language in Robotic Affordances
>
> [7] Skill induction and planning with latent language
>
> [8] Text2Motion: From Natural Language Instructions to Feasible Plans
>
> [9] TACO: Learning Task Decomposition via Temporal Alignment for Control
>
> [10] LEAGUE: Guided Skill Learning and Abstraction for Long-Horizon Manipulation
>
> [11] Learning Rational Subgoals from Demonstrations and Instructions
>
> [12] Learning grounded finite-state representations from unstructured demonstrations
>
> [13] ELLA: Exploration through Learned Language Abstraction
>
> [14] Learning with Language-Guided State Abstractions

---

> > ### Comment · Reviewer_i3As · 2023-11-22
> >
> > I would like to thank the authors for their efforts to prepare the detailed explanations, figures and extra results.
> >
> > The explanations resolve the majority of my concerns and confusions if not all of them. I'm delighted to raise my rating to 8. I would suggest the authors take some time to reflect the work done during the rebuttal in the final submission. Especially, it would be nice if the authors could modify the writing carefully to:
> > - clarify on the mode transition matrix
> > - define the loss terms correctly with math
> > - clarify on the "Mode-conditioned BC"
> >
> > I look forward to reading the final version of this paper!

---

> > > ### Author Response · Authors · 2023-11-22
> > > **Robot experiments**
> > >
> > > Dear reviewer,
> > >
> > > Thank you for your kind response. We will make those edits in the camera-ready version should our paper gets accepted. Thank you for your suggestions!
> > >
> > > **[Real-world robot experiments]** We have added some real-world robot experiments that showcase how our grounding classifier can be implemented on real-world setups as well as inputing pixel inputs beyond trajectory inputs. Please find these videos on our website at https://sites.google.com/view/grounding-plans/home#h.apnj3kgovccj. Thank you for your time to review our response!

---

### Official Review · Reviewer_Mzap · 2023-10-31

**Soundness:** 3 good
**Presentation:** 3 good
**Contribution:** 3 good
**Rating:** 6
**Confidence:** 4

**Summary:**

The paper presents a method to learn mode-conditioned policy for sequential manipulation tasks. The method proceeds in four stages: 1) prompting LLMs to generate a plan that contains multiple modes as well as keypoints and features for detecting the mode, 2) gather human demonstrations, augment them with noise, and execute in the environment to obtain success/failure labels, and 3) learning a mode classifier, and 4) learning a mode-conditioned policy. The empirical results are shown on a 2D polygon domain, where an agent needs to sequentially traverse through all the modes and reach a configuration in the final mode, and on three tasks in Robosuite.

**Strengths:**

- The paper text is overall clear
- The presented idea seems to be novel
- The finding may be of general interest for the ICLR community

**Weaknesses:**

- It seems that the paper is conflated with two distinct problems: 1) generating modes and key detection features using LLMs and grounding them for sequential manipulation tasks, and 2) learning a robust mode-conditioned policy from a few demonstrations. The paper sufficiently demonstrates (2) but more evidence needs to be shown for (1), and currently the two contributions seem quite disconnected. For example, although in the method section the paper discusses how LLMs may generate reasonable mode families and key detection features for those modes, in experiment section the experiments use manually-defined modes instead of LLM-generated modes and use manually-labeled features instead of relying on automatic mechanism for grounding. Therefore, it is questionable whether the paper should be made relevant to LLMs at all.
- The clarity of the figures need to be greatly improved. (Figure 1) It’s unclear what the task even is and what the different colored lines represent, and this has to be inferred by the reader. (Figure 3) What does the coloring mean in the center figure? How can we interpret the feasibility matrix? (Figure 4) Again, the readers may need to guess what the task is. (Figure 5) In what order of the color should the agent follows? What does the difference in coloring between each sub-figure mean here?

**Questions:**

- How are the keypoints and features being grounded in the demonstrations?
- How is the feasibility matrix generated from LLMs and is it being kept fixed for each task?

---

> ### Author Response · Authors · 2023-11-20
> **Rebuttal (1/4)**
>
> Dear reviewer Mzap,
>
> Thanks for your detailed review and feedback! Please find our response below:
>
> **[Relevance to LLM]** The major concern of the reviewer is whether LLM is integral to our framework. The reviewer acknowledges that we show sufficiently the robustness of our mode-based policy to external perturbations, but argues that we do not show enough evidence that we solve the grounding problem. We respectfully disagree because solving the grounding problem is a prerequisite for our approach to robustify policy. In the following we first clarify what we mean by grounding, and second we explain LLM's relevance both in terms of the intent and implementation of this work. Lastly, we present more evidence on how not using LLM will adversely affect how well the grounding can be learned.
>
> **[What grounding problem are we solving?]** There are at least three kinds of grounding mentioned in the literature: **1. Task grounding** - using language [1] or multi-modal tokens [2] as inputs to an imitation policy to specify tasks/goals. **2. Symbolic grounding** - predicting the boolean values of symbolic state (e.g. In(can, gripper)=True, On(marbles, spoon)=False, etc) [3, 4, 5] **3. Action grounding** - mapping LLM plan to predefined primitive actions [6, 7, 8] In our work, by grounding we do not mean task grounding, but rather symbolic grounding, where we learn classifiers that map continuous states/observations to discrete modes proposed by LLM. Since we assume each mode is associated with a single policy that’s learned from segmented demonstrations, action grounding can also be achieved as a by-product of learning the classifier, which maps the LLM planned mode sequence to a sequence of policy rollouts. We have created a **new figure -- Fig 7** that we include in the revised paper and is currently on the website, [linked here](https://sites.google.com/view/grounding-plans/home#h.xly3b8ysna28), which illustrates our overall method including a visual description of how we ground the LLM knowledge in modes.
>
> **[The intent is to enable LLM-based discrete planning]** To order to use LLM for high-level planning, existing work [6, 8] typically assume symbolic classifiers and/or action primitives are given, i.e. the symbolic/action grounding part has been manually engineered. What’s left for this top-down approach is to search for a sequence of actions (grounded in language) with high overall feasibility/success rate when applying these actions in the physical space. In contrast, **the intent of our work is to reduce human involvement in engineering the classifiers and the grounded action primitives when using LLM for discrete planning**. Specifically, we take the bottom-up approach to discover action primitives grounded in demonstrations by learning a classifier to map continuous demonstrations to discrete LLM-proposed mode sequence. Since these skills are segmented from successful demonstrations and grounded in modes/physical space, our framework do not need humans to define them a priori or calculating the feasibility of executing them in the physical space as required in the top-down approach. Note that we do not claim to completely automate humans out of the loop as our framework do require humans to provide a few demonstrations as well as prompting the LLM to generate task-relevant features and feasibility matrix. But we do believe that LLM is best used for discrete planning in the symbolic space and there is a role for humans to provide the continuous signals grounded in the physical space (rather than prompting LLM to produce the continuous components such as control signals). Our work enables this view by learning the mapping from continuous physical space to the discrete symbolic space.

---

> ### Author Response · Authors · 2023-11-20
> **Rebuttal (2/4)**
>
> **[The intent requires solving the robust policy learning task to evaluate grounding]** The reviewer might be wondering why we tackle the problem of learning a robust mode-conditioned policy from a few demonstration if the intent is to learn a grounding classifier. To evaluate the utility of grounding in the physical spaces, we need to probe the boundary of the learned classifier. While visualizing boundaries is easy in the 2d polygon domain, it is difficult in the high-dimensional manipulation space. For manipulation, mode families are a useful construct to help achieve planning success [9, 10]. The reason that mode families have boundaries is that otherwise the the motion planning won’t guarantee success. Therefore, **evaluating the effectiveness of classifier's prediction boundary can be proxied by evaluating whether the classified modes can increase the execution success rate** especially under external perturbations. The consideration of external perturbations is distinct from prior work that use pre-defined high-level actions such as “walk to \<PLACE\>” [11], “(pick ball)” [12] “open(obj)” [7]. These high-level actions are sufficient if there are no external perturbations to derail the execution. Otherwise, decomposing these actions further into manipulation modes grounded in the physical space are necessary for (1) replanning/robustifying the actions against adversarial perturbations as well as (2) explaining why some but not all perturbations will cause execution failures. Inspired by this idea, we devise a fully differentiable end-to-end explanation pipeline that predicts if a perturbed trajectory is successful or not. Only when the grounding classifier in the pipeline has learned the correct mode partitions, can the overall pipeline differentiating all successful trajectories from failure trajectories. Our explanation-based learning approach is similar to analysis-by-synthesis in other domains. For example, in NeRF only when an accurate 3d representation has been learned can the fully differentiable volumetric rendering pipeline generate images that match groundtruth from all views.
>
> **[The implementation requires LLM's common sense knowledge]** The main novel contribution is the method with which we learn the grounding classifier rather than how we implement the mode-based policy that improves robustness to perturbations. Our implementation requires LLM to provide common sense knowledge about the discrete task structure that are complementary to low-level continuous demonstrations. Specifically, (1) LLM informs how many modes there are as well as generating a matrix that describes the feasibility of transitioning from one mode to another. Without knowing the number of modes, trajectory clustering or segmentation is a NP-hard problem [13, 14]. (2) LLM reduce the dimensionality of feature space and improve data efficiency as separately investigated in [15, 16]. For example, the state of distractor objects is not useful for learning a classifier that detects different modes in the demonstrations of picking up a can object. Including distractor objects' states as inputs require significantly more counterfactual data to learn a classifier that does not pay attention to distractor objects. (3) LLM is integral to replanning at test time when there is perturbation. The utility of the learned grounding operator lies in its ability to explain when perturbation derails a plan and its capability to map the replanned discrete mode sequence from LLM to continuous policy rollout. We have created a **new figure -- Fig 8** that we include in the revised paper and is currently on the website, [linked here](https://sites.google.com/view/grounding-plans/home#h.7l0jx2g9td4d) that illustrates both how we use LLMs to generate the classifier state and how this information is used downstream to compute the classifier loss.
>
> **[Evidence that we use LLM knowledge to successfully learn grounding]** We present the following results on an anonymous project website https://sites.google.com/view/grounding-plans/home.
> 1. To show we have successfully learned grounding in the 2d polygon domain, we compare our learned mode classifications with the groundtruth on demonstration trajectories (Therefore the reporting scores are trajectory segmentation accuracies). The following table shows the comparison with ablation model (no counterfactual data) and a simple trajectory segmentation baseline by KMeans++ clustering.
>
>
> | Mode Classifier      | 3-Mode | 4-Mode | 5-Mode |
> | ----------- | ----------- | ----------- | ----------- |
> | Ours      | **0.990**       | **0.967** | **0.970** |
> | No Counterfactual Data   | 0.604  | 0.464 | 0.831 |
> | Trajectory Segmentation Baseline | 0.644 |0.554 | 0.641
>
> Additionally, in Table 1 of the paper, the MMLP-Stable\(p\) and MMLP-Stable rows show that we can achieve near 100% success rate of reaching the final mode either with or without perturbations through planning in the learned mode partitions.

---

> ### Author Response · Authors · 2023-11-20
> **Rebuttal (3/4)**
>
> 2. To show we have successfully learned grounding in the robosuite environments, we show figures on the website of segmenting the trajectories into modes similar to the groundtruth that we manually designed. We also report (below and on the website) the average mode classification accuracy (compared to ground truth) for our method. Additionally, we show videos for each robosuite task the performance of BC baseline without and with perturbations and how our mode-based imitation policy can better recover from perturbations as indirect evidence that we have learned useful mode classification.
>
> | Mode Classifier      | Can | Lift | Square Peg
> | ----------- | ----------- | ----------- | ----------- |
> | Ours (LLM-reduced State Space)     | **0.83**       | **0.83** | **0.67** |
> | Full State Space | 0.55 | 0.70 | 0.57 |
> | Trajectory Segmentation Baseline  | 0.66        | 0.56 | 0.54 |
>
>
> 3. To show the importance of prompting the LLM for the correct feasibility matrix, we run 2d polygon experiments where the 3-mode and 4-mode tasks are given a generic sequential 5-mode feasibility matrix $F^5$ and the 5-mode task is given a 3-mode feasibility matrix $F^3$. We show the resulting learned mode boundaries does not match the groundtruth. Taking the second mode in the 3-mode task (the first polygon) as an example, while our model recovers 0.946 of the first mode region (measured as the F1 score), the baseline completely mis-identified the first mode (F1 = 0). This will cause issues for robot following mode sequences when they try to recover from perturbations. See also [this section](https://sites.google.com/view/grounding-plans/home#h.12fce10e77be29fa_3) on our website for qualitative evaluations.
> 4. To show the importance of prompting the LLM to reduce the feature set, we run the default full set of features as the state representation to learn the grounding classifier for robosuite tasks. In the above table, we also provide the mode classification accuracy for this method compared to ground truth and show that for all three robosuite tasks, it is lower than our method with the LLM-generated features.
> 5. We generate feasibility matrices by querying LLMs to generate pairwise connectivities between modes (see our response to "How is the feasibility matrix generated from LLMs and is it being kept fixed for each task" below for details). Similarly, prompt LLMs to select a subset of features for training mode classifiers and policies. We include the prompts and LLM responses for robosuite tasks on our website.
>
> We hope these results on the website are sufficient evidence to show that our approach has learned grounding and how LLM is integral in this process. One can argue that our method does not need LLM as a human can easily generate the feasibility matrix or the reduced feature set for a task. But this argument exactly supports why LLM is a good proxy for a human model as they can do the same task interchangeably. And just like the other LLM-based embodied AI works [6, 7, 8, 11, 12, 15, 16], we are using LLM to reduce human work that humans are good at rather than the other way around.
>
> **[Whether we use manually labelled modes and features to learn the grounding]** There is a misunderstanding that the reviewer thinks we "use manually-defined modes instead of LLM-generated modes and use manually-labeled features instead of relying on automatic mechanism for grounding". In fact, we only use manually-defined modes and features to construct a ground-truth mode classifier, which is used to evaluate our mode classifier *learned* using LLM-generated feasibility matrix and subset features as described above in figure 7 and 8.
>
> **[Improve figure readability]**
> We have made several changes regarding figures in our updated manuscript. First, we introduced two new figures (Fig 7/8) that provide a more clear high-level explanation of the overall method as well as how the LLM is used in the grounding. You can view these figures at the top of our new website (https://sites.google.com/view/grounding-plans/home). These are aimed to replace the existing Figure 1 and Figure 3 where the reviewer had concerns around interpretability of the Figures. For Figures 4/5, we have updated figure captions per the reviewer's request in the new paper draft. We have included version of the Figures with updated captions on the website (very bottom) and plan to make further visual changes to Figure 4 to improve interpretability.
>
> **[How are the keypoints and features being grounded in the demonstrations]** In robosuite environments, the demonstration state consists of predefined object states corresponding to those keypoints shown in figure 8 (a). We add a full list of available keypoints to the prompt when query LLM to find a subset of features relevant to a task. More prompting examples can be found on the website in [this section](https://sites.google.com/view/grounding-plans/home#h.743770ss0gzs).

---

> ### Author Response · Authors · 2023-11-20
> **Rebuttal (4/4)**
>
> **[How is the feasibility matrix generated from LLMs and is it being kept fixed for each task?]** To generate the feasibility matrix, we first prompt LLM to generate the connectivity between each pair of modes. Next, we define the feasibility matrix entry $F[i, j]$ as the negative distance between each pair of modes (i.e., the shortest path) if $j$ is reachable from $i$; otherwise $F[i, j]$ is zero. In the simplest case where all mode transitions form a chain (which is true for our 2d polygon and robosuite tasks), $F[i, j] = 0$ for all $j \le i + 1$ (i.e., at mode $i$ we can transit to the next mode $i+1$ or to any previous modes $j < i$); and $F[i, j] < 0$ for all transitions that skips modes (e.g., directly make a transition from mode 1 to 3). After generating the feasibility matrix based on the task description (e.g., lift up a block from the table), we fix it for the task. The feasibility matrix is interpretable; therefore, human experts can also modify this matrix manually.
>
> [1] Language Conditioned Imitation Learning over Unstructured Data
>
> [2] VIMA: General Robot Manipulation with Multimodal Prompts
>
> [3] Grounding Predicates through Actions
>
> [4] From Skills to Symbols: Learning Symbolic Representations for Abstract High-Level Planning
>
> [5] Learning Temporal Logic Formulas from Suboptimal Demonstrations: Theory and Experiments
>
> [6] SayCan: Grounding Language in Robotic Affordances
>
> [7] Skill induction and planning with latent language
>
> [8] Text2Motion: From Natural Language Instructions to Feasible Plans
>
> [9] Multi-Modal Motion Planning in Non-Expansive Spaces
>
> [10] Integrated Task and Motion Planning
>
> [11] Language Models as Zero-Shot Planners: Extracting Actionable Knowledge for Embodied Agents
>
> [12] PDDL PLANNING WITH PRETRAINED LARGE LANGUAGE MODELS
>
> [13] NP-hardness of Euclidean sum-of-squares clustering
>
> [14] Segmentation of Trajectories on Nonmonotone Criteria
>
> [15] ELLA: Exploration through Learned Language Abstraction
>
> [16] Learning with Language-Guided State Abstractions

---

> > ### Author Response · Authors · 2023-11-22
> > **Additional evidence on how we learned grounding classifier**
> >
> > Dear reviewer,
> >
> > **[Real-world robot experiments]** We have added some real-world robot experiments that showcase how our grounding classifier can be implemented on real-world setups as well as inputing pixel inputs beyond trajectory inputs. Please find these videos on our website at https://sites.google.com/view/grounding-plans/home#h.apnj3kgovccj. Thank you for your time to review our response!

---

> > > ### Comment · Reviewer_Mzap · 2023-11-23
> > > **Thank you for the response**
> > >
> > > I would like to thank the authors for their detailed response, additional experiments, new illustrative figures, and real-world experiments. These new results and clarifications have cleared most of my previous concerns and misunderstandings of the paper. I'm happy to raise my recommendation to 6.
> > >
> > > While the paper presents an interesting and insightful use of LLMs for manipulation tasks, the reason I'm withholding a stronger recommendation is that: because LLMs can serve as good human proxy, they are often most useful when the method can be interpreted as "in either the training or deployment stage, it effectively acts as humans to reduce actual human engineering". However, it doesn't seem like the paper falls into either category.
> > >
> > > (a) In training stage,  in order to accomplish a new task (either high-level or low-level), the LLM is only used to automate a very small portion of the pipeline, because after the LLM does inference, humans are still required to provide demonstrations, which arguably involves nontrivial effort. With this amount of effort, the human can easily just write the "ground-truth" mode decomposition and transition matrix as well.
> > >
> > > (b) In deployment stage, the authors claimed that "LLM is integral to replanning at test time when there is perturbation". However, the way I understand it is that LLMs are *not* being used in closed loop for replanning. Instead, it is only used in the beginning to decompose the modes and the associated predicates/symbols. After this, it is not being queried at test time, and these mode classifiers are being used for replanning. Can authors clarify this? Regarding this point, the authors should also discuss relevance to works in Task and Motion Planning (TAMP) domain, because the symbolic predicate and the transition matrix effectively encode the "pre-conditions" and "post-conditions" in TAMP. As shown in [1] (which is also cited in the paper), after the LLM outputs a domain specification, one may use search-based planner to perform closed-loop planning, which handles external disturbances well and can accomplish similar tasks.
> > >
> > > Due to the above reasons, while the use of LLMs here is innovative, I believe it dilutes the contributions of learning robust mode-conditioned policies. If the purpose of the paper was to focus more on the LLM aspect, it would be more ideal to demonstrate on a lot larger set of tasks, as the main advantage of LLMs is that they provide generalization to diverse scenarios.
> > >
> > > Having said these, I believe the paper contains useful insights for the community and agree with other reviewers that this is a novel step towards using LLMs for robotics tasks. I also appreciate the authors' efforts during the rebuttal stage to include the new results and improve the paper. Therefore, I'm now leaning toward accepting the paper.
> > >
> > > [1] Liu, Bo, et al. "Llm+ p: Empowering large language models with optimal planning proficiency." arXiv preprint arXiv:2304.11477 (2023).

---

> > > > ### Author Response · Authors · 2023-11-23
> > > > **Final response**
> > > >
> > > > Dear reviewer,
> > > >
> > > > Thank you for your comments and we would like to respond to your concern of insufficient use of LLM in both the training and deployment stage.
> > > >
> > > > (a) In training stage, the reviewer is concerned that the use of human demonstrations dilutes the contribution of LLM. While we agree that LLM is a good human proxy, we don't think LLM is good human proxy for continuous aspect of a task as it is for discrete aspect of a task. In particular, we don't think prompting LLM to generate control signals is a principled way of doing continuous control. Our paper hopes to advocate a view that for hybrid systems (which is typical for most of long-horizon manipulations tasks), the best use of LLM is in the discrete decision-making domain. To translate the discrete decision-making by LLM to the continuous domain inevitably requries a grounding classifier that maps the continuous domain to the discrete domain. To address the reviewer's concern that the effort required for humans to generate demonstrations justifies asking humans to design feasibility matrix along the way, we want to stress that we only use a few humans demonstrations (fewer than 20), which is not a significant burden for humans, and the majority of the trajectories are generated through synthetic perturbations without humans' involvement. Without our method of using human demonstrations, manual engineering of the classifiers are required to leverage LLM discrete plan, which might be even more effort for humans. Lastly, designing feasibility matrix based on manipulation modes is non-trivial for complex tasks, and it cannot be assumed that any humans, especially those without the knowledge of mode family, can design the correct feasibility matrix.
> > > >
> > > > (b) In deployment stage, the reviewer is concerned that LLM is not being used. In fact, LLM is actually being used to plan or replan at the discrete level. This is both shown through our figure 7 (f) and the fact that our simulation and real-world robot system can replan on the fly when perturbations derail an original LLM plan. The use of our learned classifier is to continuously monitor the continuous states of the system and communicate the corresponding discrete modes the system is undergoing to the LLM for any possible replan on the fly. We will make these points clearer in the final draft and discuss relevant TAMP literature. Lastly, to address the reviewer's sugguestion of using search-based planning, we want to stress this is precisely the benefit/motivation of recovering mode boundaries--that is we can do closed-loop planning that guarantees robustness to perturbations. We also show the suggested planning performance with our learned modes in the 2d Polygon domain as detailed on our website. However, for tasks without known dynamics model to plan with (e.g. the robosuite tasks and our real-world tasks), we can only perform closed-loop planning at the discrte level by LLM. The continuous execution requires imitating demonstrations in this case.

---

> > > > > ### Comment · Reviewer_Mzap · 2023-11-23
> > > > > **Thank you for the clarification**
> > > > >
> > > > > Thank you for the clarification.
> > > > >
> > > > > Regarding (b), I do not fully understand why LLMs are queried again in the deployment stage. Since the LLMs have already outputted the sequence of modes and the transition feasibility matrix (and we have learned the mode classifier), we can simply use the mode classifier and the transition matrix for closed-loop replanning -- no LLM call is necessary. This was what I meant by "deployment stage". Is that correct?

---

> > > > > > ### Author Response · Authors · 2023-11-23
> > > > > > **New response**
> > > > > >
> > > > > > Dear reviewer
> > > > > >
> > > > > > The feasibility matrix describes all valid and invalid transitions, that is there are multiple 0 entries for the valid ones. Hence it is necessary to query the LLM to figure out which valid transition to take. Hope this helps!

---

### Official Review · Reviewer_Rhgd · 2023-11-01

**Soundness:** 3 good
**Presentation:** 3 good
**Contribution:** 3 good
**Rating:** 8
**Confidence:** 3

**Summary:**

The central idea is to leverage mode families (defines a specific type of motion constraint among a set of objects and can be chained together to represent complex manipulation behaviors) from manipulation research, in combination with LLMs for robust and grounded manipulation.  The authors framework Manipulation Modes from Language Plans (MMLP)  learns manipulation mode families from language plans and counterfactual perturbations. Given a small number of human demonstrations for the task and a small language description of the task, MMLP automatically reconstructs the sequence of mode switches required to achieve the task, and learns classifiers and control policies for each individual mode families. MMLP has four stages: prompting LLMs with a short task description to create a multi-step physical plan; generating a vast amount of counterfactual perturbed trajectories based on a small set of successful demonstrations, and subsequently, learning a classifier for each mode family outlined in the plan; and using the learned classifiers to segment all trajectories and derive mode-specific policies. Evaluation is shown using a simple 2D continuous-space domain and Robosuite (Zhu et al., 2020) and a simulated robot manipulation environment.

**Strengths:**

- In the sea of prompt-based planning approaches for embodied AI, this work felt creative and novel. MMLP seems to combine best of many worlds e.g. capabilities of LLMs for generating abstract/high-level plans, mode families form manipulation research to ground these plans into motion constraints, and idea of counterfactuals to efficiently learn these mode families and corresponding policies.
- The use of mode families enables MMLP to be interpretable and robustly generalizable.

**Weaknesses:**

- Well-thought out analysis but weak evaluation: The evaluation seemed a bit on the weaker side.
    - Even though I liked the systematic comparison with BC-based baselines and ablations on loss analysis, the two eval domains are both really simple and in simulation. The comparison with BC really helps in validating the use of mode families, however, the evaluation doesn't seem to provide me an understanding of how MMLP would compare with other SOTA manipulation planning approaches such text2motion, vima etc. To that end, I think addition of a more broader set of baselines would strengthen the paper.
    - Real-world eval or eval with more complex tasks is also highly encouraged in the same vain. Currently I don’t have any intuition or understanding about how well would MMLP work in real world. I’d love to understand how MMLP would deal with partial observability as well. Would the addition of a “search mode” for finding the right object in clutter for instance, make MMLP brittle/ineffective?
- Opensourcing plans? The authors do not talk about releasing their code, which is important for reproducibility. I encourage the authors to consider and comment on this in their rebuttal.
- The authors highlight how prompting the LLM to find a suitable state representation for learning the classifier requires skill. It seems like this is the case for generation of perturbations as well. It is not clear to me if same style of perturbations would work for more complex manipulation setting.

**Questions:**

- The loss terms were a bit difficult to parse at the first go — I’d love a visualization if possible to better understand the loss terms.
- While it was clear that the LLM generated sub-plans were used to identify number of modes etc., it wasnt clear to me how exactly were additional things obtained from prompting LLM in the first stage of MMLP used as state in the later stages of MMLP. Could the authors explain this more clearly perhaps by adding an example of what “s” looks like for the classifiers?
- Unclear how the method would work without a simulator to generate success labels for counterfactuals. Is the availability of sim an assumption for MMLP?

---

> ### Author Response · Authors · 2023-11-20
> **Rebuttal (1/1)**
>
> Dear reviewer Rhgd,
>
> Thanks for your detailed review and feedback! Please find our response below:
>
> **[Comparison with LLM-based planning methods such as text2motion/VIMA]** Both text2motion and VIMA solve a different grounding problem than ours and thus would not be the best baselines. Specifically, there are at least three kinds of grounding mentioned in the literature: **1. Task grounding** - using language [1] or multi-modal tokens [2] as inputs to an imitation policy to specify tasks/goals. **2. Symbolic grounding** - predicting the boolean values of symbolic state (e.g., In(can, gripper)=True, On(marbles, spoon)=False, etc) [3, 4, 5] **3. Action grounding** - mapping LLM plan to predefined primitive actions [6, 7, 8] Text2Motion deals with action grounding and VIMA deals with task grounding. In contrast, our work solves symbolic grounding, where we learn classifiers that map continuous states/observations to discrete modes proposed by LLM. Since we assume each mode is associated with a single policy that’s learned from segmented demonstrations, action grounding can also be achieved as a by-product of learning the classifier, which maps the LLM planned mode sequence to a sequence of policy rollouts. That being said, we agree with reviewer's point of adding more baselines, so we are adding a baseline that achieves symbolic grounding through clustering in the 2d polygon domain (see the ablation studies for 2D polygon, linked [here on the website](https://sites.google.com/view/grounding-plans/home#h.8wpp2pzbin0p)) and similarity-based trajectory segmentation in the robosuite domain (see the mode classification comparison table and examples for the new baseline, [here on the website](https://sites.google.com/view/grounding-plans/home#h.7l0jx2g9td4d)). To clarify the grounding, we have created a **new figure: Fig 7** that we include in the revised paper and is currently on the website.
>
> **[Visualization of loss function and state $s$ for classifier]**
> We have created a **new figure -- Fig 8** that we include in the revised paper and is currently on the website in [this section](https://sites.google.com/view/grounding-plans/home#h.pmzm9p2g4j7v) that illustrates both how we use LLMs to generate the classifier state and how this information is used downstream to compute the classifier loss.
>
> **[Does our approach require a simulator to generate success labels?]** Having a simulator perhaps is required for methods that require per-step dense labeling. Since our approach only requires sparse label at the end of a trajectory, engineering a task success classifier for real-world tasks is not infeasible and hence simulators are not required. For example, for a scooping task where the goal is to transport marbles from one bowl to another bowl, one can engineer a classifier to detect if the perturbed execution still manages to transport at least one marble to the goal location (i.e., the second bowl).
>
> **[How does MMLP work in the real-world setting?]** We are preparing real-world experiments and will show results on the website. https://sites.google.com/view/grounding-plans/home.
>
> **[Opensourcing plans?]** Yes we are opensourcing the code soon!
>
> **[Will the current perturbations work for more complex tasks?]** While figuring out the best perturbations strategies for different tasks with different complexity is not the main focus of this paper, we agree designing perturbations that are sufficient to generate counterfactual outcomes is a requirement for our method to work. In future work, we will investigate how to prompt LLM to generate different perturbation strategies for each task.
>
> [1] Language Conditioned Imitation Learning over Unstructured Data
>
> [2] VIMA: General Robot Manipulation with Multimodal Prompts
>
> [3] Grounding Predicates through Actions
>
> [4] From Skills to Symbols: Learning Symbolic Representations for Abstract High-Level Planning
>
> [5] Learning Temporal Logic Formulas from Suboptimal Demonstrations: Theory and Experiments
>
> [6] SayCan: Grounding Language in Robotic Affordances
>
> [7] Skill induction and planning with latent language
>
> [8] Text2Motion: From Natural Language Instructions to Feasible Plans

---

> > ### Comment · Reviewer_Rhgd · 2023-11-22
> > **Post rebuttal**
> >
> > Thanks for the clarifications, additional figures, and robot experiments.
> >
> > Given that authors are focused on symbol grounding problem, it would be great if they can provide comparisons with other approaches for symbol grounding [3,4] in their final version.
> >
> > I look forward to reading the final version!

---

> ### Author Response · Authors · 2023-11-22
> **Robot experiments**
>
> Dear reviewer,
>
> **[Real-world robot experiments]** We have added some real-world robot experiments that showcase how our grounding classifier can be implemented on real-world setups as well as inputing pixel inputs beyond trajectory inputs. Please find these videos on our website at https://sites.google.com/view/grounding-plans/home#h.apnj3kgovccj. Thank you for your time to review our response!

---

### Meta-Review · Area_Chair_3tFB · 2023-12-09

**Metareview:**

This paper proposes a new approach for sequential manipulation tasks by leveraging LLMs to generate plans over keypoints. Experiments are conducted on a variety of problems from Robosuite and a 2D polygon domain. The reviewers find this work overwhelmingly creative and novel. The writing is clear and convincing. A few concerns on the environment were mentioned. The author's rebuttal discussed real robot results that were convincing. Overall, a solid paper that could be improved with incorporating the new late-breaking robot results and discussion.

**Justification For Why Not Higher Score:**

The paper as is has good experiments, but not strong enough for consideration as Oral or paper award.

**Justification For Why Not Lower Score:**

This is a solid paper with a novel idea and strong experimental results.

---

### Decision · Program_Chairs · 2024-01-16

Accept (spotlight)